# Visual-TCAV: Concept-based Attribution and Saliency Maps for Post-hoc Explainability in Image Classification

**Antonio De Santis**\*                 *antonio.desantis@polimi.it*
*Politecnico di Milano*

**Riccardo Campi**\*                  *riccardo.campi@polimi.it*
*Politecnico di Milano*

**Matteo Bianchi**                   *matteo.bianchi@polimi.it*
*Politecnico di Milano*

**Marco Brambilla**                  *marco.brambilla@polimi.it*
*Politecnico di Milano*

**Reviewed on OpenReview:** *https://openreview.net/forum?id=SLhOOW5rhu*

## Abstract

Convolutional Neural Networks (CNNs) have shown remarkable performance in image classification. However, interpreting their predictions is challenging due to the size and complexity of these models. State-of-the-art saliency methods generate local explanations highlighting the area in the input image where a class is identified but cannot explain how a concept of interest contributes to the prediction. On the other hand, concept-based methods, such as TCAV, provide insights into how sensitive the network is to a human-defined concept but cannot compute its attribution in a specific prediction nor show its location within the input image. We introduce Visual-TCAV, a novel explainability framework aiming to bridge the gap between these methods by providing both local and global explanations. Visual-TCAV uses Concept Activation Vectors (CAVs) to generate class-agnostic saliency maps that show where the network recognizes a certain concept. Moreover, it can estimate the attribution of these concepts to the output of any class using a generalization of Integrated Gradients. We evaluate the method's faithfulness via a controlled experiment where the ground truth for explanations is known, showing better ground truth alignment than TCAV. Our code is available at `https://github.com/DataSciencePolimi/Visual-TCAV`.

## 1 Introduction

As the performance of deep learning models has grown significantly over recent years, their complexity has also increased, making it difficult for users to understand how these models make decisions. As a result, they are often referred to as *black-box* models, since only their inputs and outputs are known, while their internal mechanisms are too complex for humans to comprehend. This lack of algorithmic transparency (von Eschenbach, 2021) has been shown to reduce trust in AI-based systems (Lipton, 2016), particularly in critical fields such as healthcare or autonomous driving in which neural networks are becoming increasingly employed (Turay & Vladimirova, 2022; Cai et al., 2020). Additionally, debugging models becomes challenging without comprehending the process they use to make predictions. To this end, the field of Explainable AI (XAI) has made significant progress in developing techniques for explaining black-box models (De Santis et al., 2025). However, determining whether a certain human-understandable concept is recognized by the network and how it influences the prediction remains a significant challenge. In computer vision, widely used

---

\*Equal contribution.

explainability approaches use saliency maps to localize where a class is identified in an input image, but they can't explain which high-level features led the model to its prediction. For instance, these methods cannot determine whether a golf ball was recognized by the spherical shape, the dimples, or some other feature. To address this, Kim et al. (2018) introduced TCAV (Testing with Concept Activation Vectors), a method that can discern whether a user-defined concept (e.g., dimples, spherical) correlates positively with a selected class. However, TCAV cannot measure the importance of a concept in a specific prediction or show the locations within the input images where the concepts are recognized. Since global concept scores summarize the model's behavior by averaging over many inputs, they may not necessarily reflect the model's rationale for each individual instance. This is particularly important when investigating mispredictions, especially in critical domains such as medical imaging or autonomous driving, where analysts need to verify what led the model to a specific prediction and whether it was spatially grounded.

In this article, we introduce a novel explainability framework, namely Visual-TCAV, integrating the core principles of both saliency and concept-based methods to overcome their respective limitations. This framework can be applied to any layer of a CNN whose output is a set of feature maps. We design Visual-TCAV to satisfy three main desiderata for concept-based explanations: (D1) *spatial grounding*, by providing concept maps that localize where the network recognizes concepts of interest; (D2) *per-instance attribution*, by estimating how much each concept contributes to a class output for a given input; and (D3) *aggregatability*, by supporting the aggregation of per-instance attributions across inputs into global, class-level explanations.

## 2 Related Work

For image classification, early methods primarily provide explanations via saliency maps that highlight the most important regions in the input image to predict a certain class. A model-agnostic approach for generating such visualizations involves studying the model's input-output relationship by creating a set of perturbed versions of the input and analyzing how the output changes with each perturbation. Examples of this paradigm include LIME (Ribeiro et al., 2016), which uses random perturbations, and SHAP (Lundberg & Lee, 2017), which estimates the importance of each pixel using Shapley values. A different approach that instead tries to access the internal workings of the model was originally proposed by Simonyan et al. (2014) and consists of generating saliency maps based on the gradients of the model output w.r.t. the input images. This idea led many researchers (Springenberg et al., 2014; Smilkov et al., 2017) to investigate how to exploit gradients to produce more accurate saliency maps. Among them, Selvaraju et al. (2017) proposed Grad-CAM, a generalization of CAM (Zhou et al., 2016) which extracts the gradients of the logits (i.e., raw pre-softmax predictions) w.r.t. the feature maps and uses a Global Average Pooling (GAP) operation to transform these gradients into class-specific weights for each feature map. It then performs a weighted sum of these feature maps to produce a class localization map. However, Sundararajan et al. (2017) demonstrated that gradients can saturate, leading to an inaccurate assessment of feature importance. To address this issue, they introduced Integrated Gradients (IG), an axiomatic attribution method that works by integrating the gradients along a path from a baseline (e.g., a black image) to the actual input image, providing fine-grained saliency maps via per-pixel attribution.

Saliency methods provide per-instance, pixel-level explanations and class localization. They are therefore limited in connecting predictions to high-level semantic concepts and in supporting aggregation across inputs. To overcome these limitations, Kim et al. (2018) proposed TCAV, a method that investigates the correlations between user-defined concepts and the network's predictions using a set of example images representing a concept. For instance, example images of the concept "striped" can be used to determine whether the network is sensitive to this concept for predicting the "zebra" class. This is accomplished by learning a Concept Activation Vector (CAV), which is a vector orthogonal to the decision boundary of a linear classifier trained to differentiate between the feature maps of concept examples and random images. From this, a sensitivity score is computed using the signs of the dot products between the CAV and the gradients of a selected class w.r.t. the feature maps. TCAV has also been used to analyze concept-level biases and can be considered complementary to saliency maps. Indeed, while Grad-CAM and IG apply exclusively to individual predictions, TCAV provides only global explanations. However, TCAV does not show where concepts are identified within the input images, making it challenging to assess whether a high score can truly be attributed to the intended concept. Moreover, TCAV computes the network's sensitivity to a concept,

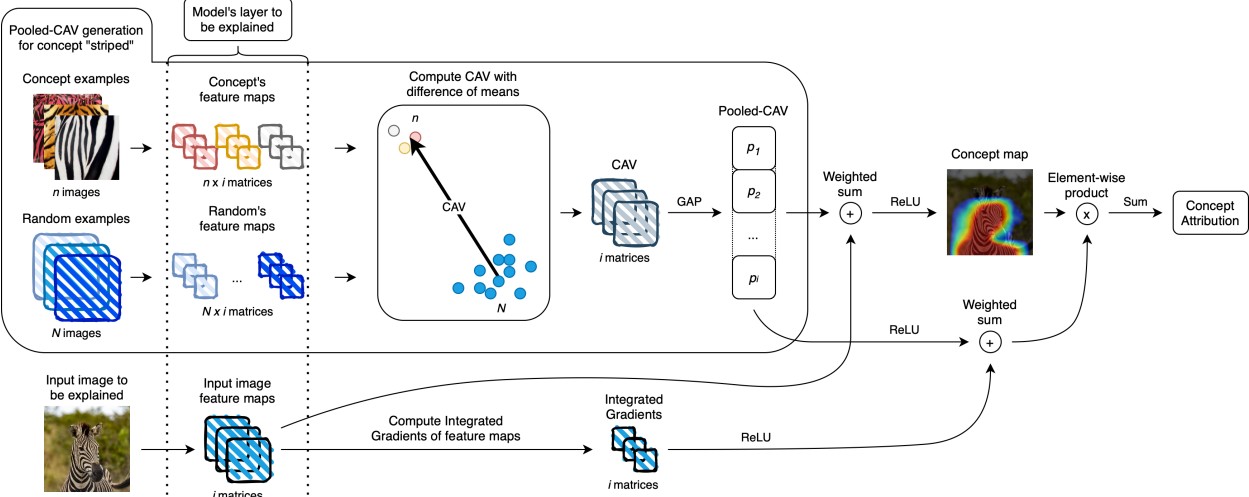

Figure 1: The Visual-TCAV pipeline for generating local explanations. A pooled-CAV is computed using the feature maps of user-defined concept examples and random images. A concept map is then produced through a weighted sum of the pooled-CAV and the image's feature maps. Finally, a concept attribution is obtained by extracting the IG attributions of the neurons that the concept activates using the pooled-CAV and the concept map, which is used as a spatial mask.

but not the magnitude of its importance, as the score only depends on the signs of the directional derivatives. For instance, the concepts "grass" and "dimples" might have identical TCAV scores for the "golf ball" class, even if one contributes substantially more to the prediction.

TCAV has received attention within the XAI community, leading to various applications (Lucieri et al., 2020; Cai et al., 2019) and extensions (Crabbé & van der Schaar, 2022; Graziani et al., 2018). A notable extension that tries to add localization for a concept of interest is CAVLI (Shukla et al., 2023), which uses LIME to decompose images into superpixels and uses CAVs to assess their similarity to the concepts. In parallel, several unsupervised approaches have also been proposed (Ghorbani et al., 2019; Zhang et al., 2021; Fel et al., 2023b; Akula et al., 2020) to discover concepts automatically. These methods typically involve cropping images of a class to generate patches, which are then re-inserted into the network, and activations are clustered, resulting in a set of extracted concepts. More recently, Sparse Autoencoders (SAEs) have also shown promising results in discovering interpretable features across both vision (De Santis et al., 2026; Fel et al., 2023a) and language domains (Bricken et al., 2023; Bereska & Gavves, 2024). However, it is often the case that concepts of interest are not being extracted even though they are important for the task (Sharkey et al., 2025). For the scope of this paper, we will focus only on supervised scenarios in which the concept to study is manually defined by the user.

## 3  Visual-TCAV

This section presents Visual-TCAV, whose methodology is outlined in Fig. 1. Local explanations can be generated for any layer and consist of two key components: (i) the *concept map*, a saliency map that serves as a visual representation of where the network recognized the concept in the input image and (ii) the *concept attribution*, a numerical value that estimates the importance of the concept for a selected class. To derive global explanations, the process is replicated across multiple input images and the concept attributions are averaged to quantify how the concept influences the network's decisions across a wide range of inputs.

### 3.1  CAV Generation and Spatial Pooling

Following the original TCAV framework, the initial step of our method consists of computing a Concept Activation Vector (CAV) from a set of example images representing a user-defined concept and a set of

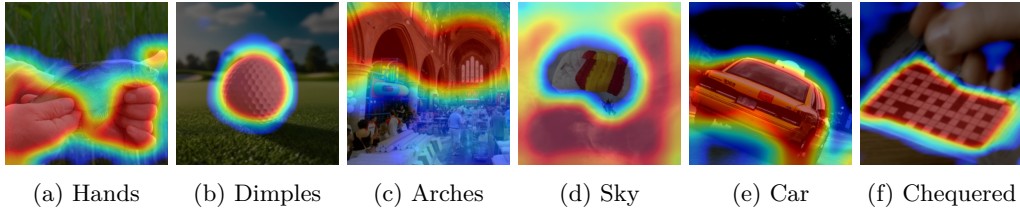

(a) Hands     (b) Dimples     (c) Arches     (d) Sky     (e) Car     (f) Chequered

Figure 2: Examples of class-independent concept maps for various input images and concepts.

negative examples (e.g., random images). Specifically, we use the *Difference of Means* method proposed by Martin & Weller (2019) to compute the CAV, as they demonstrated that this approach produces more consistent CAVs than logistic classifiers or SVMs. As the name suggests, this method uses the arithmetic mean to determine the centroids of both the concept's activations and the activations of random images. A CAV is then computed directly as the difference between the concept centroid and the random centroid. This is also mathematically equivalent to the approach proposed by Pahde et al. (2025). Since we are interested in identifying which feature maps are activated by the concept, irrespective of its spatial location within the example images, we apply a GAP operation on the obtained CAV. The result is a vector of scalar values whose length is equal to the number of feature maps of the layer under consideration. Each element of this vector is associated with a feature map, and its raw value approximates the degree of correlation between that feature map and the concept. Moving forward, we refer to this vector as the *pooled-CAV*.

### 3.2 Concept Map

Using the pooled-CAV, we derive a concept map $\boldsymbol{M}^c$ that spatially localizes a concept $c$ within the input image by applying Eq. (1). Specifically, we compute a weighted sum of the feature maps $\mathbf{F}$ from the selected layer, where each channel $i$ is weighted by the corresponding element $p_i^c$ of the pooled-CAV. Subsequently, we apply a ReLU function to retain only the regions positively correlated with the concept.

$$\boldsymbol{M}^c = ReLU\left(\sum_i p_i^c \cdot \mathbf{F}_{:,:,i}\right) \tag{1}$$

To enable meaningful comparisons of concept activation across different input images, we apply a custom min-max normalization to the concept map based on a predefined range, as shown in Eq. (2). The normalization range boundaries ($\ell^c$ and $u^c$) are derived from the example images provided by the user to represent the concept. Specifically, the upper bound $u^c$ is computed by first generating concept maps from positive example images $\mathcal{D}_{pos}$, then calculating the median of their contraharmonic means (*chmean*), which serve to heuristically extract values of high activation within each concept map. The lower bound $\ell^c$ is computed using the same method but with the concept maps of negative example images $\mathcal{D}_{neg}$ instead. The normalized concept map $\hat{\boldsymbol{M}}^c$ is then obtained by clipping any value outside this predefined range and then scaling it to $[0, 1]$. The notation $[\boldsymbol{A}]_a^b$ denotes clipping each element of $\boldsymbol{A}$ within the interval $[a, b]$, i.e., $[\boldsymbol{A}]_a^b = min(max(\boldsymbol{A}, a), b)$, where the *min* and *max* are applied element-wise. An ablation study on different normalization approaches is provided in Appendix H.

$$\hat{\boldsymbol{M}}^c = \frac{[\boldsymbol{M}^c]_{\ell^c}^{u^c} - \ell^c}{u^c - \ell^c}, \quad \text{where} \quad \begin{cases} u^c = median\left(chmean_{i \in \mathcal{D}_{pos}}(\mathbf{M}_{:,:,i}^c)\right) \\ \ell^c = median\left(chmean_{i \in \mathcal{D}_{neg}}(\mathbf{M}_{:,:,i}^c)\right) \end{cases} \tag{2}$$

By overlaying the normalized concept map on the input image, we generate a class-independent visualization that highlights the region of the image where the network recognized the concept (examples are shown in Fig. 2). This allows us to know, for any input image, the concept's location and its degree of activation w.r.t. an ideal concept defined by the user. Additionally, concept maps act as a qualitative validation for whether the CAV represents the intended concept, without requiring activation maximization techniques (Mordvintsev et al., 2015) or sorting images based on their similarity to the CAV.

### 3.3 Concept Attribution

We aim to gain insights into the network's decision-making process by estimating the importance of a given concept for a target prediction. Concretely, for a chosen layer and target class, we first compute attribution scores for the layer's activations with respect to the target class logit, and then use the pooled-CAV to weight those attributions according to how strongly each feature map is associated with the concept.

**Integrated Gradients in feature-map space.** We compute feature maps attributions using Integrated Gradients (IG), but applied to the activation space of a chosen layer rather than to the input pixels. Let $\mathbf{F}$ denote the feature maps at the chosen layer for a given input image, and let $\mathbf{F}_0$ denote the baseline feature maps (in our case, an all-zero tensor with the same shape as $\mathbf{F}$). We denote by $z_k(\mathbf{F})$ the logit (pre-softmax score) of class $k$ obtained by running the model forward from the chosen layer while setting that layer's feature maps to $\mathbf{F}$. We consider a straight-line path in the feature maps space from $\mathbf{F}_0$ to $\mathbf{F}$, and compute the integrated gradients of the target class logit with respect to the feature maps along this path. As shown in Eq. (3), we use the IG definition adapted from Sundararajan et al. (2017) to operate in feature map space. Since IG satisfies the completeness axiom, the attributions sum to the difference between the target class's logit and the baseline's logit for that class, regardless of which layer is used as input. Given a target class $k$, we refer to attributions w.r.t. feature maps of a specific input image as $\mathbf{IG}^k$.

$$\mathbf{IG}^k = (\mathbf{F} - \mathbf{F}_0) \odot \int_{\alpha=0}^{1} \frac{\partial\, z_k(\mathbf{F}_0 + \alpha(\mathbf{F} - \mathbf{F}_0))}{\partial \mathbf{F}}\, d\alpha. \tag{3}$$

In practice, this integral is approximated numerically by evaluating gradients at a finite number of steps along the interpolation path using the Riemann trapezoidal rule.

**Normalizing attribution magnitude.** Since the scale of these attributions is the same as the raw logits, making interpretation difficult, we normalize the attributions so that their sum is between 0 and 1. We retain only the positive attributions, as we focus on evidence supporting the target class[1]. We then compute the difference between the logits of the input and those of the baseline, followed by a ReLU and $[0, 1]$ scaling as shown in Eqs. (4a) and (4b), where $\mathbf{z}(\mathbf{F})$ denotes the vector of logits obtained by running the model forward from the chosen layer while setting that layer's feature maps to $\mathbf{F}$.

$$\Delta \boldsymbol{z} = ReLU\left(\boldsymbol{z}(\mathbf{F}) - \boldsymbol{z}(\mathbf{F}_0)\right) \tag{4a} \qquad\qquad \Delta \hat{\boldsymbol{z}} = \frac{\Delta \boldsymbol{z}}{max(\Delta \boldsymbol{z})} \tag{4b}$$

The attributions for the target class $(k)$ are then scaled so that their sum equals the normalized difference $\Delta \hat{z}_k$ for that class, as shown in Eq. (5a).

$$\hat{\mathbf{IG}}^k = \frac{ReLU(\mathbf{IG}^k)}{\sum ReLU(\mathbf{IG}^k)} \cdot \Delta \hat{z}_k \tag{5a} \qquad\qquad \hat{\boldsymbol{p}}^c = \frac{ReLU(\boldsymbol{p}^c)}{max(\boldsymbol{p}^c)} \tag{5b}$$

**Weighting attributions by concept alignment.** To estimate the attribution of a concept $c$ for a target class $k$, we utilize the pooled-CAV as a per-channel vector of weights to perform a weighted sum of the normalized attributions $\hat{\mathbf{IG}}^k$ toward that target class. Before weighting, we apply a ReLU and max normalization to the pooled-CAV, as shown in Eq. (5b), to progressively down-weight feature maps that are less aligned with the concept, and emphasize those that are strongly concept-related. We refer to the normalized pooled-CAV as $\hat{\boldsymbol{p}}^c$. The rationale behind using the ReLU here is to keep only feature maps whose activation is positively aligned with the concept.

**Combining attribution, concept weighting, and spatial masking.** As shown in Eq. (6), we compute the concept attribution $a^{c,k}$ by first aggregating the normalized attributions $\hat{\mathbf{IG}}^k$ across feature maps using the normalized pooled-CAV weights $\hat{p}_i^c$, obtaining a single spatial attribution map for the target class. We then

---

[1]For binary classifiers with a single output head, we treat the negative class score as $-z$. For the chosen target class (either $z$ or $-z$), we keep only positive IG values and normalize by their sum as in Eq. (5a).

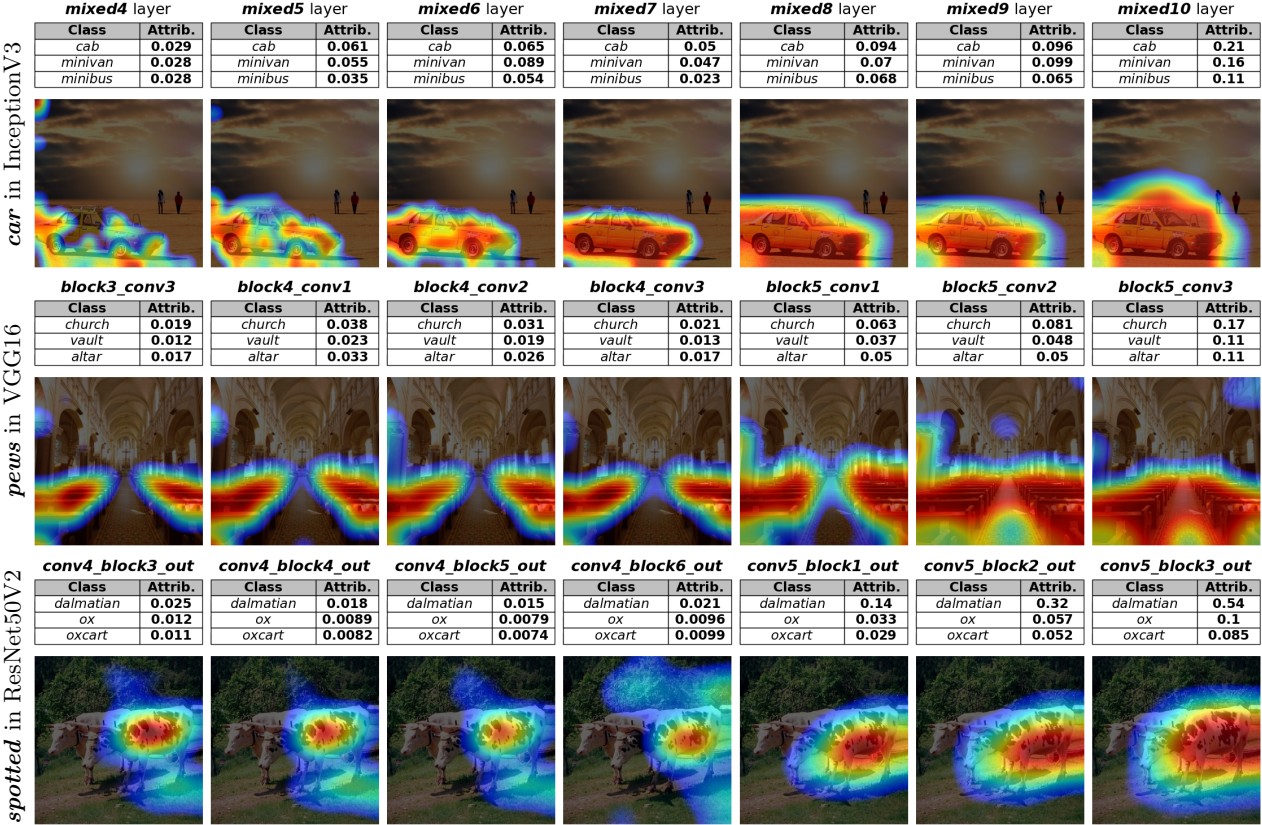

Figure 3: Layer-wise local explanations for various concepts and architectures. We compute the attribution of each concept for the top three predicted classes, ranked by class probability, and the last seven layers.

multiply this map element-wise by the normalized concept map $\hat{\boldsymbol{M}}^c$, which acts as a spatial mask. Finally, we sum over all spatial locations to obtain a scalar score that quantifies the attribution of concept-aligned activations to the target class prediction.

$$a^{c,k} = \sum \left( \hat{\boldsymbol{M}}^c \odot \sum_i (\hat{p}_i^c \cdot \mathsf{I}\hat{\mathsf{G}}_{:,:,i}^k) \right) \tag{6}$$

**Interpretation and global aggregation.** The concept attribution is a per-concept metric, meaning that two concepts can have significantly different attributions even if they are recognized in the same location of the input image, resulting in similar concept maps. Since concepts may naturally overlap in the same spatial regions, attributions are not meant to form a spatial partition of the image. This is natural in CNNs, where multiple channels may activate in the same region while detecting different patterns. Furthermore, since concepts can be correlated or hierarchically related (e.g., "car" and "wheel"), concept attributions are not assumed to be additive, and attributions should not be interpreted as a decomposition of the prediction that can be summed across concepts. Finally, to obtain a global explanation, we can average the concept attribution across multiple input images to measure the overall importance of a concept for a selected class. For instance, we can calculate a global attribution of the "striped" concept for the "zebra" target class by averaging the attribution of "striped" across a large number (e.g., 200) of images containing zebras.

## 4    Results

This section presents the results of applying Visual-TCAV to a range of widely used CNNs. We show that our method (i) can reveal how concepts contribute to mispredictions, (ii) provides insights into where concepts are detected within the network, and (iii) aligns with ground truth in controlled experiments.

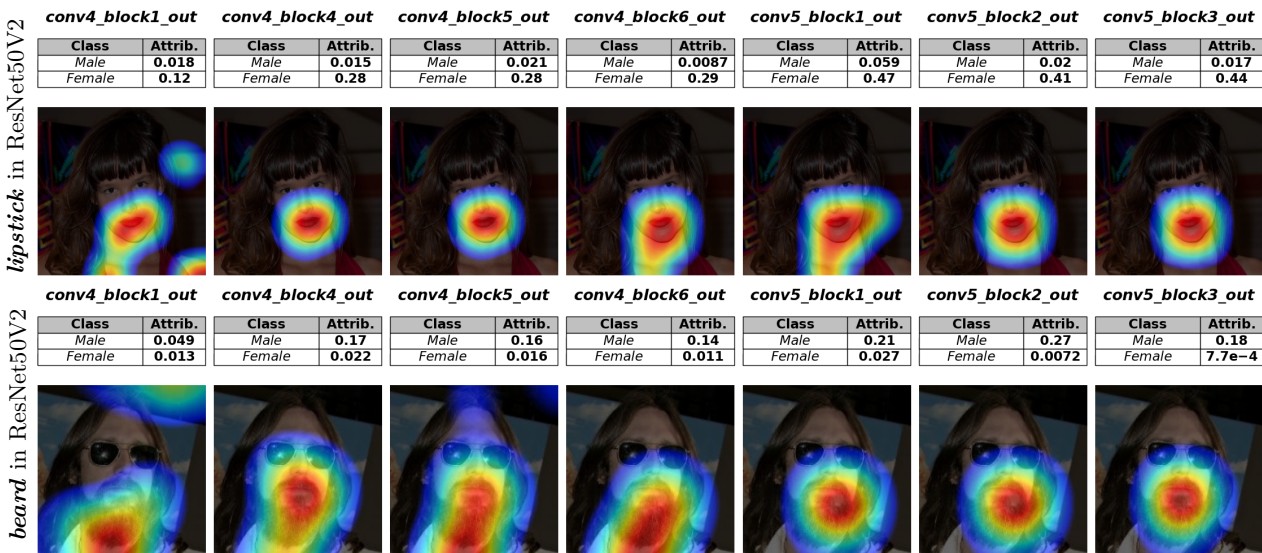

Figure 4: Layer-wise local explanations for a ResNet50V2 (He et al., 2016) model trained on the CelebA (Liu et al., 2015) dataset for celebrity gender classification. The concepts tested are "lipstick" and "beard", both significantly contributing to the prediction.

## 4.1 Experimental Setup

Our experiments are conducted on ResNet50V2 (He et al., 2016), InceptionV3 (Szegedy et al., 2016), and VGG16 (Simonyan & Zisserman, 2015), all pre-trained on the ImageNet (Deng et al., 2009) dataset, as well as a ResNet50V2 model trained from scratch (i.e., without ImageNet weights) on the CelebA (Liu et al., 2015) dataset for celebrity gender classification. The tested concepts were sourced from the Describable Textures Dataset (DTD) (Cimpoi et al., 2014), obtained from popular image search engines, or generated with Stable Diffusion (Rombach et al., 2022) (more on this in Appendix E). For concepts collected via search engines, we use 30 example images per concept, which can be considered a reasonable amount, as shown in the stability experiment provided in Appendix G. For DTD concepts, we use all 120 available images, and for generated concepts, we use 200 images. For ImageNet negative examples, we follow the approach recommended by Martin & Weller (2019), selecting 500 random images. In the case of CelebA, images containing the concept are used as positive examples, while those without it serve as negative examples. Regarding the computation of the integrated gradients, we used 300 steps, which are usually enough to approximate the integral within a 5% error margin (Sundararajan et al., 2017). Experiments are conducted on an Intel i7 13700k with an RTX 4060Ti 16GB and 32 GB of RAM, using TensorFlow 2.15.1, CUDA 12.2, and Python 3.11.5. With this setup, local explanations for 7 layers take 1 to 2 minutes, while global ones with 200 class images and 7 layers take from 5 to 30 minutes depending on the model. For global explanations, the computation time remains nearly constant regardless of the number of concepts processed simultaneously.

## 4.2 Local Explanations

While concept maps are class-independent, the attribution of each concept depends on the class considered. In our examples, we examine the top three predicted classes and apply Visual-TCAV to a subset of the CNN layers. The rationale behind these layer-wise explanations is that they enable us to visualize where specific concepts are learned within the network. For instance, the "car" concept in Fig. 3 is not well recognized in earlier layers, corroborating previous findings that higher-level features are typically extracted deeper into the network (Zeiler & Fergus, 2014; Olah et al., 2017; Bau et al., 2017; Bianchi et al., 2024). On the other hand, concept maps in earlier layers are more fine-grained (see the "pews" concept in Fig. 3) due to their neurons having smaller receptive fields. Additionally, we observe a substantial increase in attributions in deeper layers, which is aligned with other studies (Morcos et al., 2018; Amjad et al., 2022) showing deeper

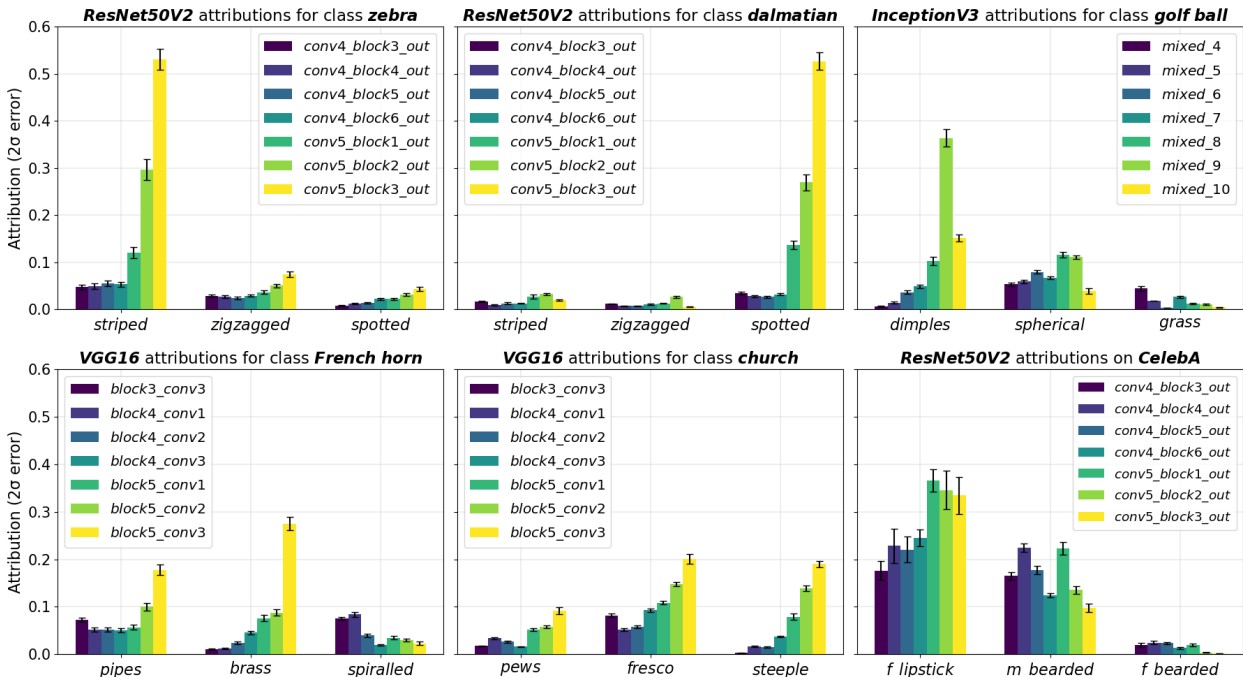

Figure 5: Global explanations for various concepts, classes, and networks. Each bar chart shows the attributions of three concepts for a specific class across the last seven layers of each network. In the CelebA dataset, "m_" denotes attribution toward the "male" class, while "f_" indicates attribution toward the "female" class. Each concept's attribution is averaged over 200 input images.

neurons exhibit higher class selectivity due to their proximity to the output. However, this is slightly less pronounced on the CelebA dataset (refer to Fig. 4), where the "lipstick" and "beard" concepts exhibit class selectivity even in earlier layers, which may be due to the lower task complexity. The utility of concept-based explanations lies mainly in their ability to show not only which image regions the network is focusing on, but also what the model recognizes in those regions and how much this contributes to each output. This can be especially useful for revealing the cause of mispredictions. For instance, the third image in Fig. 3 is an "ox" wrongly classified as "dalmatian" for which we observe that the network's decision is largely influenced by the "spotted" concept, which accounts for more than half the logit value of the "dalmatian" class.

## 4.3 Global Explanations

The concept attribution is a per-concept metric of importance; thereby, we can derive global explanations by aggregating this attribution across multiple input images of a selected class. In our experiments, we utilize 200 images per class for each global explanation. For concepts that are inherently part of the class (e.g., "striped" for "zebra", or "dimples" for "golf ball"), we can directly use any image representing that class. On the other hand, for concepts that appear sporadically, we only use images where the concept is present. For instance, we only use images of church interiors for "pews" and "fresco" concepts, and images of church exteriors for the "steeple" concept. This ensures that the explanations are independent of the frequency of the concept's appearance in the class images.

Results of global explanations for various concepts are provided in Fig. 5. The attributions match intuitive expectations, considering, for instance, the importance of "striped" for "zebra" or "spotted" for "dalmatian". However, such high attribution may suggest the model over-relies on certain concepts, using them as shortcuts for class discrimination. This may lead the network to misclassify out-of-distribution cases, such as the "ox" being incorrectly labeled as a dalmatian in Fig. 3. This may also pose issues for concepts like "lipstick" in gender classification, potentially leading to biased predictions. Furthermore, while the final layer typically

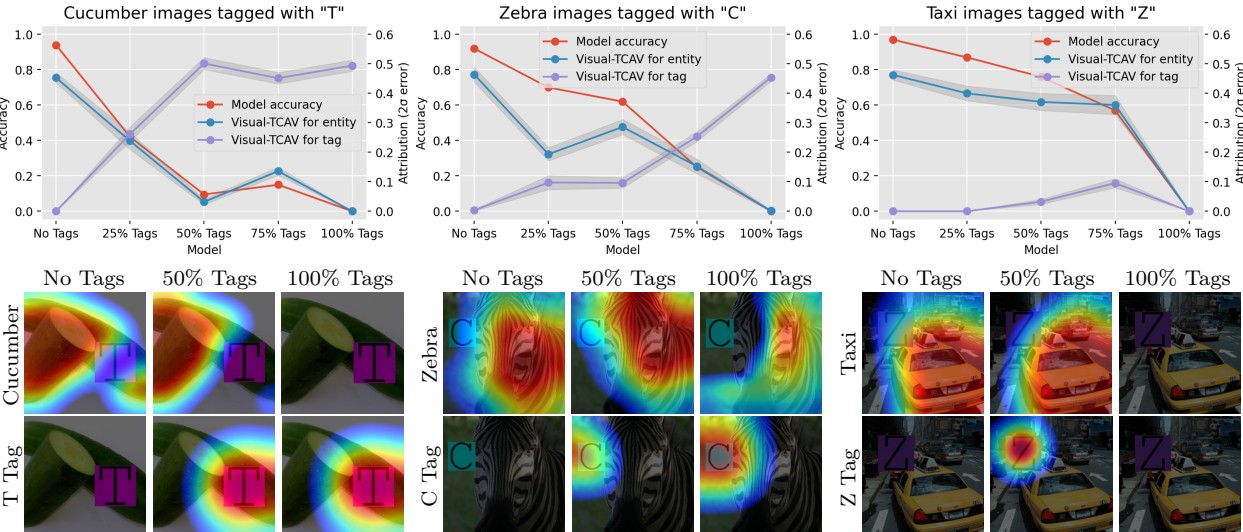

Figure 6: The results of the validation experiment. The upper section of the figure shows the models' accuracies and the Visual-TCAV attributions for both entities and tags across all models. The lower section provides examples of concept maps for the no tags model, 50% tags model, and 100% tags model.

has the highest attribution, which is expected for class discriminative concepts, there are instances, such as "dimples" and "spherical" for "golf ball", where concepts recognized in the earlier layers have a greater impact on the prediction. Empirically, we found such behavior to be dependent not only on the concept but also on the model architecture. More examples of explanations can be found in Appendices C and D.

## 4.4 Quantitative Evaluation with Ground Truth

We conducted a validation experiment to evaluate the faithfulness of Visual-TCAV. In this experiment, we train a series of convolutional networks in a controlled setting, where the ground truth for explanations is known, and assess whether the Visual-TCAV attributions match this ground truth. For this purpose, we create a dataset of three classes – cucumber, taxi, and zebra – which are the same classes used in the TCAV validation experiment. We then create multiple versions of this dataset by altering a percentage of the images with a tag, represented by a letter enclosed in a randomly sized square, added in a random location of the image. Specifically, zebra images are tagged with a "Z" in a purple square, taxi images with a "T" in a magenta square, and cucumber images with a "C" in a cyan square. From these tagged images, we create five datasets: one of images without tags, and four others with 25%, 50%, 75%, and 100% of tagged images, respectively. Each dataset is then used to train a different model, each including six convolutional layers and a GAP layer. Depending on the dataset used for training, each model may learn to recognize either the entities (i.e., cucumbers, taxis, and zebras), the tags, or both, and will decide which ones to give more importance. To obtain an approximated ground truth assessing which concept – entity or tag – is more important, we ask the models to classify a set of 200 incorrectly tagged test images per class. In this test set, taxis are tagged with the "Z", cucumbers are tagged with the "T", and zebras are tagged with the "C". If the network accuracy remains high, it indicates that the entity is more important than the tag, and thus, we expect its attribution to be higher. On the other hand, if the performance deteriorates on these wrongly tagged images, it indicates that the tag is more important than the entity. Hence, the tag's attribution should be higher. We obtain the CAVs for entities using images of each class as concept examples and the other two classes with random tags as negative examples. For tags, we use random images containing that tag as concept examples and images of cucumbers, taxis, and zebras containing the other two tags as negative examples. We use the same incorrectly tagged test set to compute the concept attributions for both entities and tags across the last convolutional layer of all models.

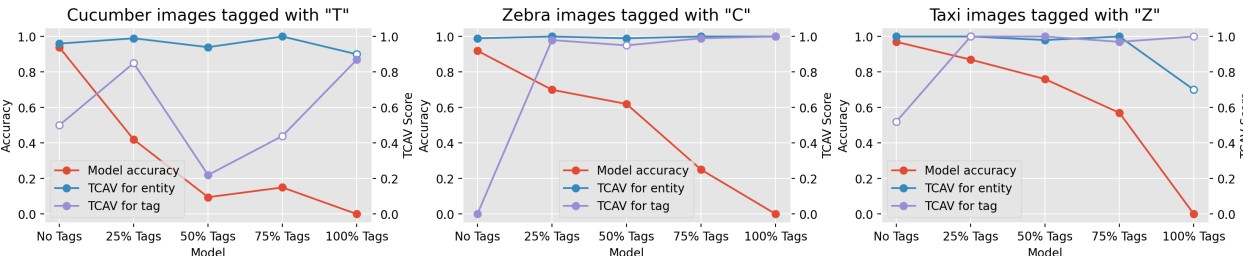

Figure 7: TCAV scores for tags and entities for models trained on different percentages of tagged images. Results not statistically significant (p-value<0.05) according to the TCAV test are indicated with white dots.

The results are shown in Fig. 6 and, as expected, an increase in the percentage of tagged images correlates with a decrease in accuracy for all classes. In particular, for the "cucumber" class, the accuracy declines much faster than other classes, with most images being incorrectly classified as taxis. This suggests that even the models trained on a small fraction of tagged images tend to overfit on the "T" tag. The concept attributions for both the "cucumber" entity and the "T" tag closely mirror this ground truth. The "zebra" entity and the "C" tag are also consistent with the ground truth: the attributions for "zebra" show a positive correlation with accuracy, whereas the attributions for the "C" tag demonstrate a clear inverse correlation. Notably, the networks did not pay much attention to the "Z" tag, focusing instead on the absence of the other two tags to classify zebras. Indeed, the model trained with 100% of images tagged classifies any image without a "C" or a "T" tag as "zebra", regardless of whether the "Z" tag is present or not. This is confirmed by our method, which assigns an attribution of nearly zero to both the "Z" tag and the "taxi" entity for the aforementioned model. We also tested saliency methods, such as Grad-CAM and IG, to further validate these findings. These methods do not highlight the "Z" tag either, but rather the entire image, in search of the "zebra" class (see Appendix B). For all models excluding the one trained with 100% of tags, the accuracy for "taxi" remains high, implying that these models are indeed capable of recognizing the "taxi" entity. The concept attribution for the "taxi" entity aligns with this observation. In the lower part of Fig. 6, we provide examples of concept maps for the models trained with a different percentage of tagged images. As expected, the model trained without tags can recognize the entities but not the tags, the model trained with 50% tagged images can recognize both, and the model trained with 100% tagged images struggles to recognize the entities but effectively identifies the "T" and "C" tags.

### 4.4.1 Comparative Analysis with TCAV

The primary difference between our concept attribution and the TCAV score is that our method considers the magnitude of the gradients, not just their direction. This allows us to measure the concept's impact on the predictions, beyond just the network's sensitivity to it. To demonstrate this, we compute the TCAV scores for tags and entities across each validation model (see Fig. 7). For the model trained without tags, the TCAV scores align with ground truth, showing high sensitivity to entities and no or negative sensitivity to tags (where 0.5 indicates no sensitivity and 0 signifies negative sensitivity). However, for the other models, TCAV struggles to capture variations in concept importance as defined by ground truth. These models yield high TCAV scores for both entities and tags, despite the decline in accuracy, indicating that their contributions to predictions differ significantly. This limitation arises because TCAV relies solely on the signs of gradients, making it incapable of distinguishing the relative importance of these concepts. Therefore, while the TCAV score may be valuable in explaining whether the network learned to correlate a certain concept with a class, our concept attribution provides a more comprehensive explanation by quantifying the extent to which a given concept influences the network's decisions. A qualitative and quantitative comparison with saliency methods (Grad-CAM and Integrated Gradients) is provided in Appendix F.

## 5 Conclusion

In this article, we presented Visual-TCAV, a novel method that provides concept-based local and global explanations for image classification models by estimating the attribution of user-defined concepts to the

network's predictions. Visual-TCAV also generates saliency maps to show where concepts are identified by the network, providing users with visual evidence that CAVs align with their intended concepts. Our method's effectiveness was demonstrated in widely used CNNs and a ground truth experiment, where it successfully identified the most important concept in each examined model.

### 5.1 Limitations and Future Work

One limitation is that collecting example images requires some effort and domain knowledge and may also introduce some subjectivity. To address this, future work could explore generating CAVs directly from text using methods such as Text-To-Concept (Moayeri et al., 2023), which aligns the activations of multi-modal models like CLIP (Radford et al., 2021) with vision-only models. Following recent work on synthetic CAVs (Campi et al., 2025), another promising direction is fine-tuning text-to-image generative models with a few real examples using DreamBooth (Ruiz et al., 2023) to generate images better aligned with the analyst's intended concept for CAV training. A further limitation is that, since concept attributions are not additive, Visual-TCAV does not provide a completeness guarantee over an arbitrary user-defined concept set, and therefore cannot verify that the tested concepts are exhaustive of what the model has learned.

As future work, we plan to investigate the applicability of our method to other tasks like regression as well as other architectures like Vision Transformers (ViTs) (Dosovitskiy et al., 2021). For ViTs in particular, a natural adaptation could operate on patch tokens to obtain concept maps. However, token mixing through self-attention implies that patch tokens can encode evidence originating from other regions, making spatial grounding less reliable compared to CNN feature maps (Jeanneret et al., 2025; Darcet et al., 2024). Since addressing this would require non-trivial design adaptations, we leave a systematic ViT evaluation to a dedicated future study. It may also be interesting to study interconnections between concepts to determine not only the concept attribution toward a class but also toward other concepts in deeper layers.

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

# A  Appendix Overview

In the appendix, we provide:

## B   IG and Grad-CAM for 100% tags model

We provide the results obtained by applying IG (Sundararajan et al., 2017) and Grad-CAM (Selvaraju et al., 2017) to the 100% tags model (see Fig. 8). These methods align with Visual-TCAV in showing that this model does not pay attention to the "Z", but rather to the absence of the "T" and the "C" for predicting the "zebra" class.

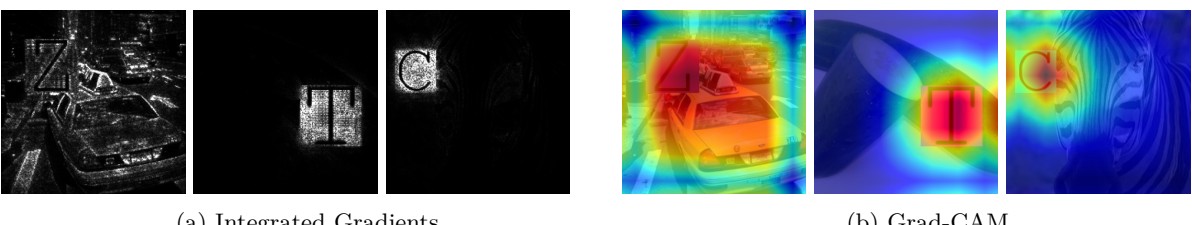

(a) Integrated Gradients                                    (b) Grad-CAM

Figure 8: IG and Grad-CAM for the model with 100% tags, searching respectively for the classes "zebra", "taxi", and "cucumber". Both methods highlight the "T" for class "taxi" and the "C" for class "cucumber", but fail to recognize the "Z" for class "zebra".

## C   Additional results of Local Explanations

Continuing from the results presented in the main paper, we further provide additional local explanations for more input images and concepts in Figs. 9 to 11.

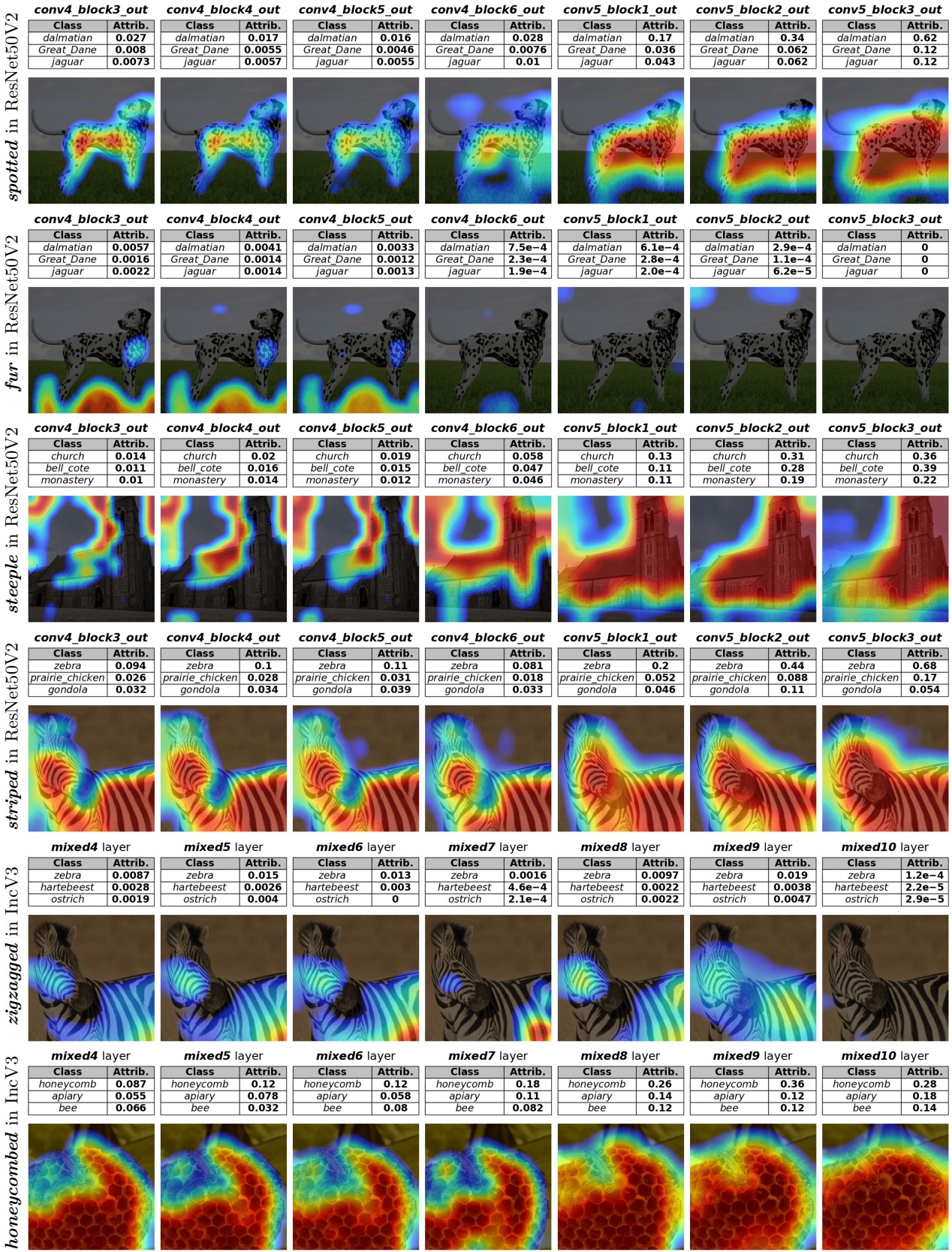

Figure 9: More examples of layer-wise local explanations for various concepts and networks. (Part 1)

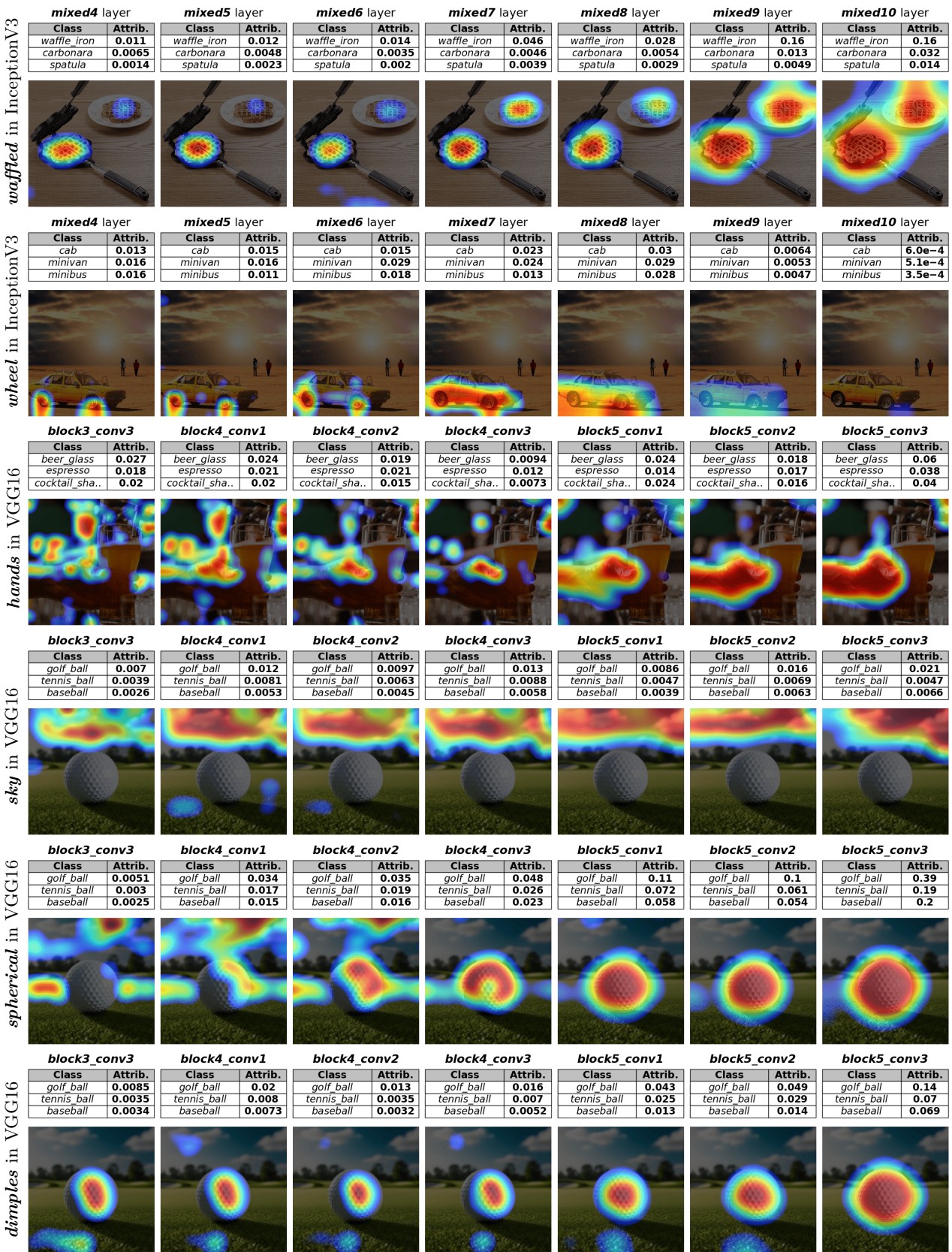

Figure 10: More examples of layer-wise local explanations for various concepts and networks. (Part 2)

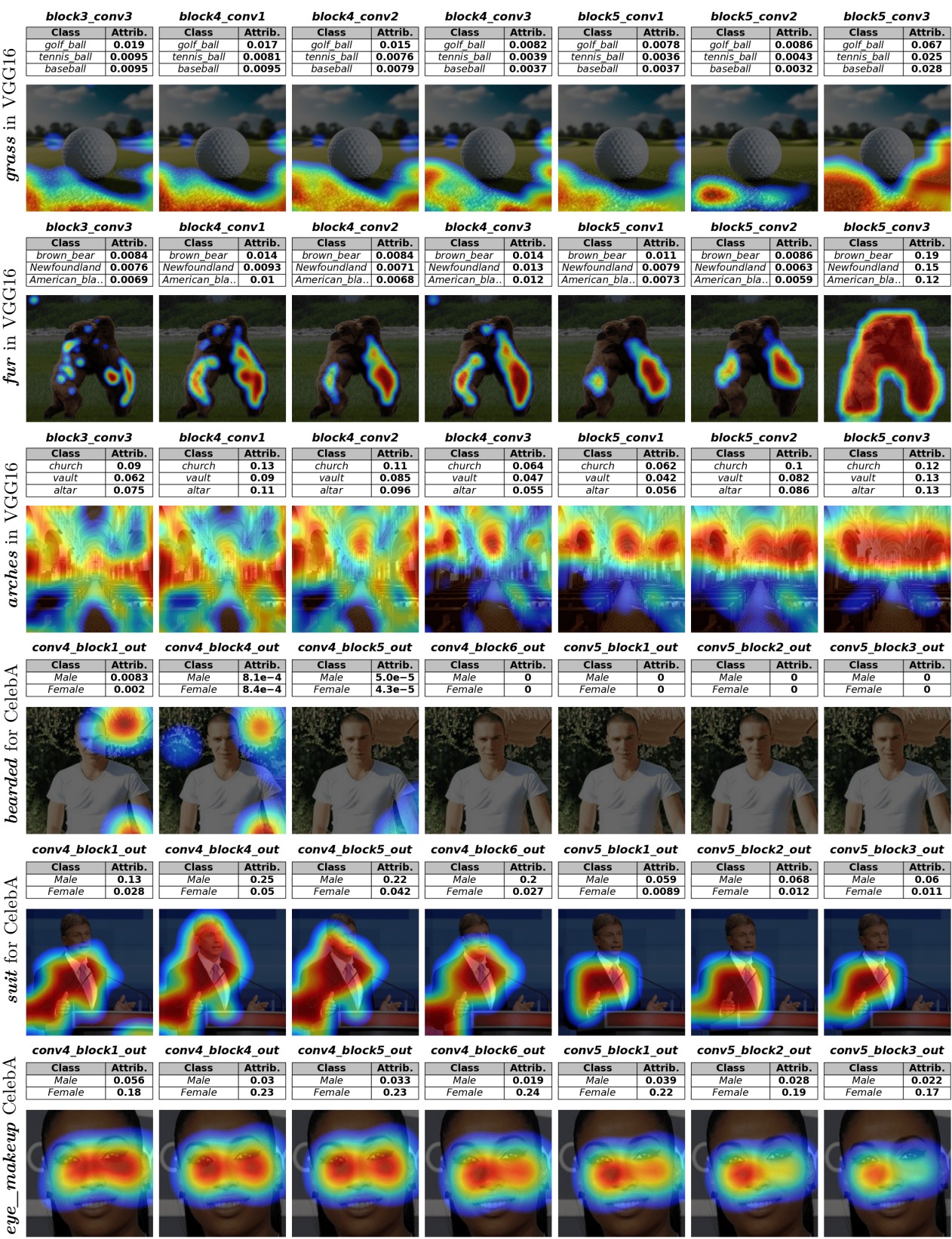

Figure 11: More examples of layer-wise local explanations for various concepts and networks. (Part 3)

# D    Additional results of Global Explanations

In Figure 12 we present additional global explanations for various classes and concepts.

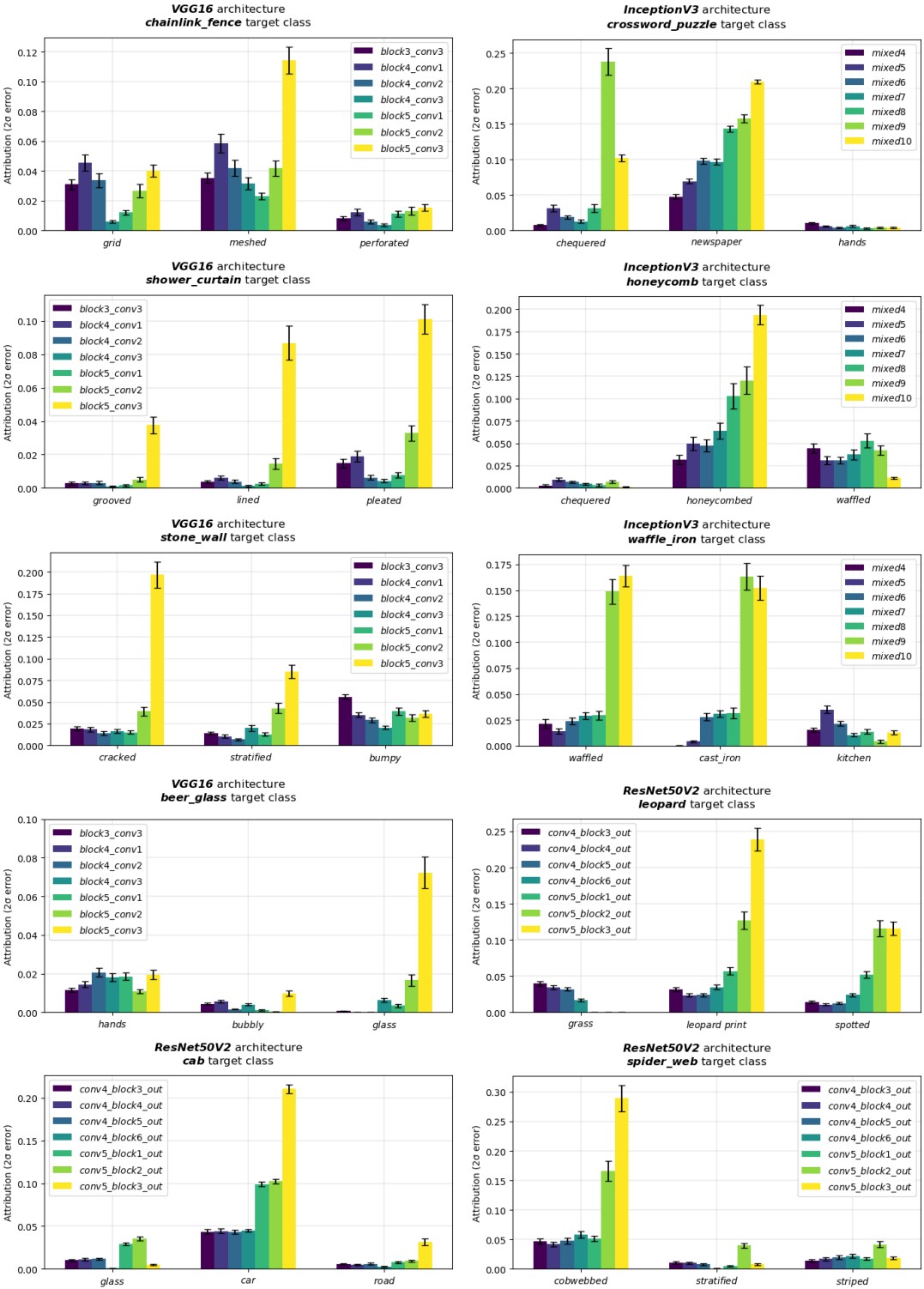

Figure 12: More examples of global explanations for various classes, concepts, and networks.

# E   Example images for generated concepts

Some of the concepts used in the paper were generated using Stable Diffusion v1.5 (Rombach et al., 2022) with default parameters. In particular, we generated the following concepts: "pews", "fresco", "arches", "sky", "pipes", and "brass". We used just the concept name as a prompt and generated 200 images per concept. A subsequent manual revision was still necessary to eliminate errors and artifacts. In Fig. 13, we provide three example images for each generated concept.

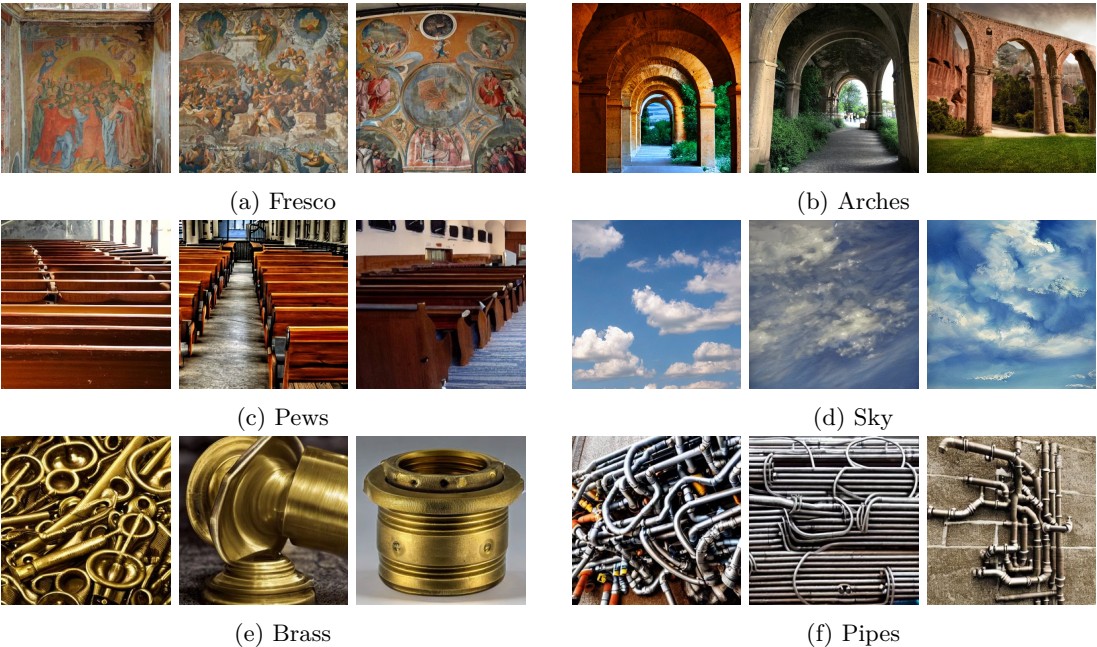

(a) Fresco  (b) Arches

(c) Pews  (d) Sky

(e) Brass  (f) Pipes

Figure 13: We provide three example images for each concept generated with Stable Diffusion v1.5.

# F   Comparison with IG and Grad-CAM

## F.1   Qualitative comparison

In Figure 14, we apply IG and Grad-CAM to some of the images analyzed in the paper, with the aim of qualitatively assessing the differences in output and scope between these methods and Visual-TCAV. As discussed in previous sections, saliency methods are among the most widely used approaches for explaining black-box image classifiers, as they enable the visualization of the most relevant regions of an image for a given prediction. However, these methods provide limited insights beyond identifying these regions. Specifically, they do not explain what features or concepts the model is identifying in those regions, how these features contribute to a specific prediction, nor can they provide global explanations.

Considering the "church (interiors)" image, IG reveals that the model primarily focuses on the pews, while Grad-CAM indicates attention to both the pews and the arches, although with lesser emphasis on the latter. These observations align with the results derived from Visual-TCAV (Figs. 3 and 11), which show that the concepts "pews" and "arches" are indeed important for the prediction, with "pews" more than "arches". Moreover, Visual-TCAV also provides a quantitative analysis of their contributions, explains "what" the network is seeing in the image in terms of the example images in Figs. 13b and 13c, and offers a global explanation by computing the average attribution of these concepts across multiple church images. Furthermore, from Visual-TCAV, we can also derive that the "arches" concept is not class discriminative in this case, as it has a similar importance for the classes "vault" and "altar".

Often, key concepts may also remain undetected by saliency methods. For example, in the case of the "golf ball" and "bear" images, saliency methods fail to reveal that the "grass" concept is somewhat relevant for the

former and that the "fur" concept is highly significant for the latter. Instead, saliency maps mainly highlight the golf ball itself, while Grad-CAM focuses on the bear's head and feet. Additionally, these methods are unable to discern whether the model recognizes the dimples on the golf ball, its spherical shape, or both, since these features visually overlap. In contrast, Visual-TCAV enables such distinctions by attributing these overlapping features to specific concepts with different importance.

Considering the saliency maps for the CelebA dataset, especially those generated by Grad-CAM, they are all very similar, making it difficult to understand whether the model learned to recognize specific attributes such as "beard", "lipstick", or "suit". However, while Visual-TCAV can provide explanations in terms of such concepts, other facial features can be very complex and are also difficult to define as human-understandable concepts, making this task challenging to interpret for any state-of-the-art XAI method.

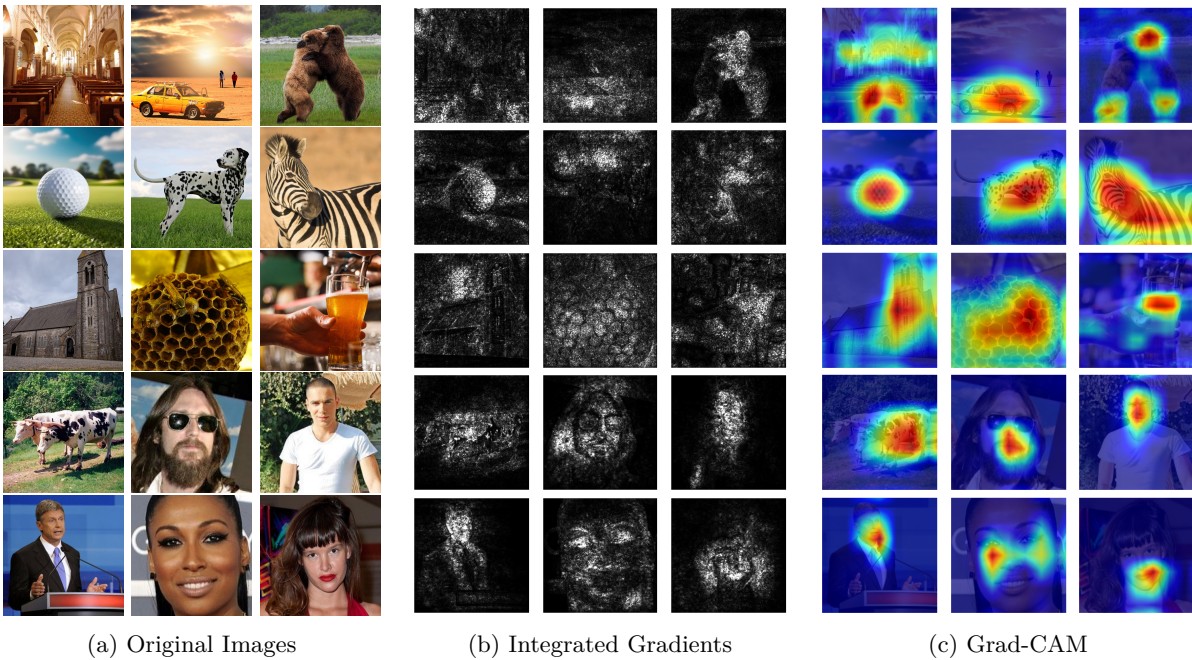

(a) Original Images  (b) Integrated Gradients  (c) Grad-CAM

Figure 14: IG and Grad-CAM for the images tested in the paper. Explanations of "church" (interiors), "bear", and "golf ball" are from VGG-16. Explanations of "church" (outside), "dalmatian", "ox", and "zebra" are from ResNet50V2. Explanations of "cab", "honeycomb", and "waffle iron" are from InceptionV3. All explanations are computed in the last convolutional layer, considering the predicted class as the target class.

## F.2 Quantitative comparison

In this section, we present the results of a quantitative experiment designed to evaluate and compare the effectiveness of Visual-TCAV, IG, and Grad-CAM in communicating which concepts are more important in a given prediction. To perform this experiment, we took the "No Tags" and "100% Tags" models from the experiment in Section 4.4 and considered the classes "cucumber" and "taxi" with their respective "C" and "T" tags, as for these classes and models, we have a clear ground truth for explanations. Indeed, from Fig. 6, we know by construction that for the "No Tags" model, the entity itself (i.e., cucumbers and taxis) is more important than the tag, while for the "100% Tags" model, the tag is more important than the entity. We selected a total of 60 images, equally split across the two classes, and generated explanations using Visual-TCAV, IG, and Grad-CAM for the two models (for a total of 360 unique explanations). Each image-explanation pair was then presented to a Multimodal Large Language Model (GPT-5 in our case), which was asked to rate, based on the explanation, which concept between the entity and the tag appeared to be more important for the model's decision. A possible answer was also "Don't know" in case, based on the explanation, it was too difficult to tell which concept was more important.

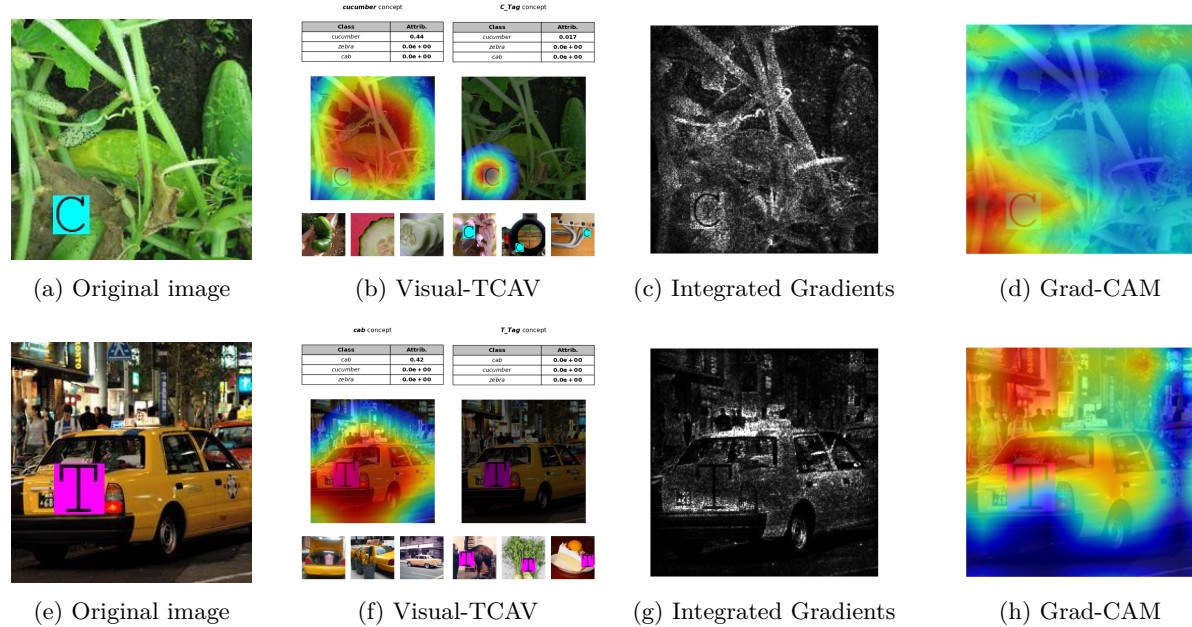

Figure 15: Explanations from Visual-TCAV, IG, and Grad-CAM for the model trained without tags. In this model, the approximated ground truth is that the model pays much more attention to the entity than the tag (which was not present in the training set). While with Visual-TCAV the explanation matches this ground truth, this is not clear from IG and Grad-CAM.

For the "No Tags" model, the ground truth label was the entity, while for the "100% Tags" model, the ground truth label was the tag. "Don't know" responses were counted as incorrect. As for the metrics considered, for each method, we computed its accuracy (i.e., fraction of correct judgment over all images) and a statistical significance applying McNemar's test between Visual-TCAV and the two baselines (Grad-CAM and IG). The results in terms of accuracy are shown in Table 1, while the results of the McNemar's tests are shown in Table 2. Fig. 15 shows two examples where saliency maps communicated the wrong concept importance. Overall, Visual-TCAV was able to communicate correctly which concept was more important 98.33% of the time, while saliency methods (Grad-CAM and Integrated Gradients) achieved substantially lower accuracies of 71.67% and 65.83%, respectively. This suggests that explanations based solely on saliency maps may be misleading, which is the same conclusion reached in a similar experiment with human subjects by Kim et al. (2018). Statistical analysis using McNemar's test confirms that these results are statistically significant with p-values $\ll 0.05$. The exact prompts used in this experiment are provided below:

```
 Visual-TCAV: You are an evaluator judging which visual concept a classifier relied on more strongly for a prediction
based on the output of an explainability method.
The first image is the input to the classifier and was predicted as class <class>.

The explainability method used is called Visual-TCAV, and given a concept, it provides a heatmap showing whether the network
recognized that concept and where, and the attribution of the concept towards each class.
The second image is the explanation for the concept <concept1>, and the third is the explanation for the concept <concept2>.
In the explanations, below the heatmap, are shown examples of images containing the concept being analyzed.
Question: Based only on these explanations, which concept appears more responsible for the model's decision?
Answer only with the name of the concept, or if you can't tell from the explanation, then answer "na".

 IG & Grad-CAM: You are an evaluator judging which visual concept a classifier relied on more strongly for a
prediction based on the output of an explainability method.
The first image is the input to the classifier and was predicted as class <class>.
The second image is the explanation.

The explainability method used is called <method>, and it provides a saliency map highlighting the most important image
region for the prediction.
Question: Based only on the explanation, which feature appears more responsible for the model's decision between <concept1>
and <concept2>?
Answer only with the name of the concept, or if you can't tell from the explanation, then answer "na".
```

Table 1: Comparison of accuracy and number of "don't know" responses across methods. Visual-TCAV achieves consistently higher accuracy than pure saliency approaches on both classes and overall.

| Class | Method | Accuracy (%) | Don't know (%) |
|---|---|---|---|
| cucumber | Visual-TCAV | 96.67 | 1.67 |
| cucumber | Grad-CAM | 60.00 | 23.33 |
| cucumber | Integrated Gradients | 51.67 | 0.00 |
| taxi | Visual-TCAV | 100.0 | 0.00 |
| taxi | Grad-CAM | 83.33 | 6.67 |
| taxi | Integrated Gradients | 80.00 | 0.00 |
| Overall | **Visual-TCAV** | **98.33** | **0.83** |
| Overall | Grad-CAM | 71.67 | 15.00 |
| Overall | Integrated Gradients | 65.83 | 0.00 |

Table 2: McNemar tests comparing Visual-TCAV against Grad-CAM and IG. $b =$ (Visual-TCAV correct, other method wrong) and $c =$ (other method correct, Visual-TCAV wrong).

| Class | Comparison | $b$ | $c$ | $p$-value |
|---|---|---|---|---|
| cucumber | Visual-TCAV vs Grad-CAM | 24 | 2 | $1.05 \times 10^{-5}$ |
| cucumber | Visual-TCAV vs IG | 27 | 0 | $1.49 \times 10^{-8}$ |
| taxi | Visual-TCAV vs Grad-CAM | 10 | 0 | $1.95 \times 10^{-3}$ |
| taxi | Visual-TCAV vs IG | 12 | 0 | $4.88 \times 10^{-4}$ |
| Overall | Visual-TCAV vs Grad-CAM | 34 | 2 | $1.94 \times 10^{-8}$ |
| Overall | Visual-TCAV vs IG | 39 | 0 | $3.64 \times 10^{-12}$ |

## G  Experiments on Visual-TCAV stability

In this section, we experiment on the stability of Visual-TCAV. Specifically, we are interested in evaluating (i) the stability of Visual-TCAV with respect to the number of concept example images used for learning the pooled-CAVs, and (ii) the robustness of concepts when perturbed with random images. To address these points, we perform two separate experiments inspired by the procedure used by Martin & Weller (2019) to assess the stability of the *difference of means* and the TCAV Score. The experiments are all performed using the final layer of the same ResNet50V2 model, ensuring comparability of the results. Regarding the example images, we select two classes, "zebra" and "church", and perform an evaluation on the "striped" and "steeple" concepts, respectively.

For the first experiment, starting from a concept defined through an arbitrary amount of training images (specifically, 120 images for "striped" and 75 images for "steeple"), we evaluate both the similarity of pooled-CAVs and the concept attribution when using a lower number of example images. To do this, we apply random sampling without replacement to the original concept images, and we apply Visual-TCAV to derive an updated pooled-CAV and the global concept attribution across the same 50 test images. We computed these metrics considering four different subsets of the concepts over 50 random runs, ultimately resulting in the plots of Fig. 16. From these plots, we observe that when considering both a texture-like concept, such as "striped", or a less homogeneous concept, such as "steeple", utilizing 30 example images instead of the original amount leads to a contained error for Visual-TCAV's concept attribution (i.e., with a mean absolute error $< 0.01$ for both concepts). At the same time, the pooled-CAVs obtained with 30 concept images present a cosine similarity very close to the original. In general, these experiments suggest that 30-50 example images for defining a concept could be a reasonable amount. However, future studies on a large variety of concepts and networks are still necessary to draw definitive conclusions. Notably, in these experiments, even just 5 images were able to provide a rough estimate, even though the risk of misrepresenting the concept is slightly higher.

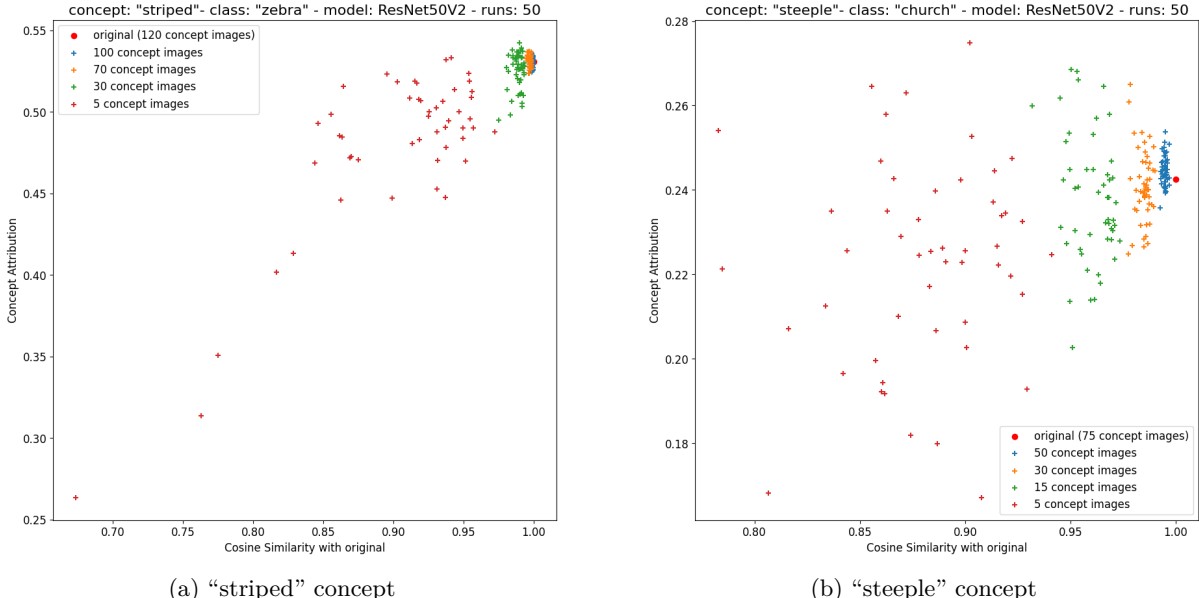

(a) "striped" concept                    (b) "steeple" concept

Figure 16: Analysis of results for "striped" and "steeple" concepts computed with a different number of example images. Each point represents the concept attribution and pooled-CAV obtained by removing n example images from a concept set.

We perform a similar experiment to test Visual-TCAV's stability when the example images contain noise (i.e., random images not correlated with the concept). The purpose is to assess how careful an analyst should be when collecting example images. To this end, we apply random sampling without replacement to select a subset of images from the "random" concept and inject them into the training images. By deriving the updated pooled-CAVs, the global concept attribution, and by repeating this procedure for an increasing number of noisy images, we can compare the deviation from the initial pooled-CAV representing the concept, and the variation of Visual-TCAV's concept attribution score. We perform this test both considering all the available example images and also with just 30 images. The results are shown in Figs. 17 and 18. From these plots, we observe that adding a few random images to the concept examples does not change the attribution significantly. The worst case is when adding 30 random images to 30 example images, in which we see a slight increase in attribution. This can be explained by the fact that we are also considering the attribution of the background, which also includes some noise accumulated by the integrated gradients. Another effect of adding many random images is that the upper bound to activate the concept map also decreases slightly, as it depends on the concept examples. Overall, from these experiments, we can conclude that results from our method are meaningful even when adding a significant amount of noise (up to 100%) to the example images. In Fig. 19, we also provide examples of concept maps computed with different amounts of example images and with added noise.

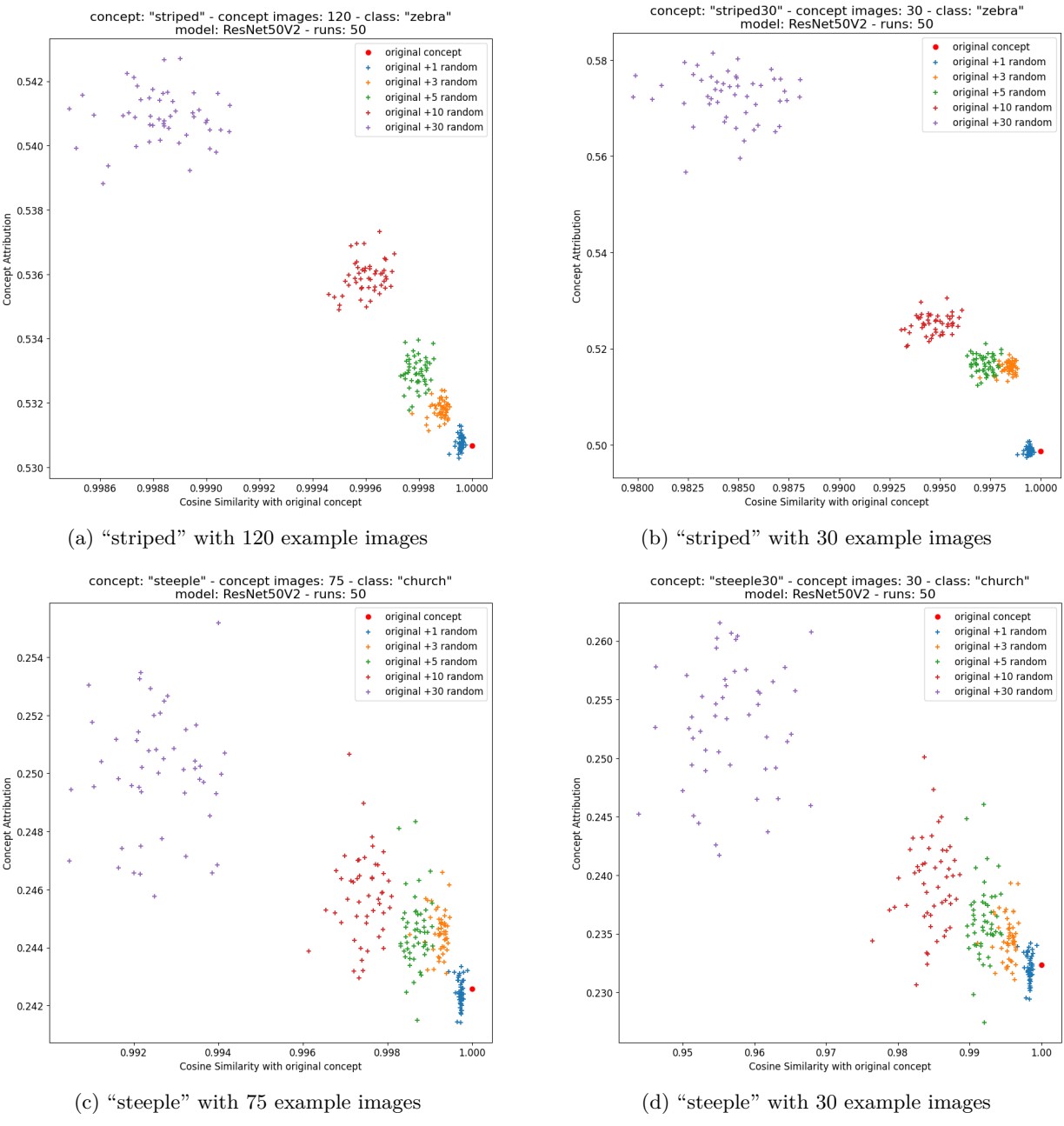

Figure 17: Analysis of results for "striped" and "steeple" concepts computed with a different number of example images and with random images added as noise to the concept set. Each point represents the concept attribution and pooled-CAV obtained by adding N random images to the example images of a concept.

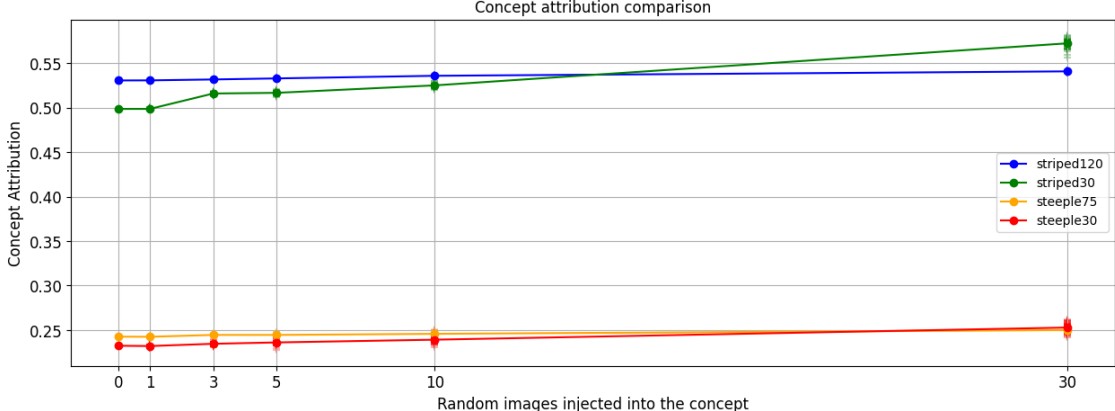

Figure 18: Summary of the concept attribution results for the second experiment. The line plots represent the variation of the average concept attribution when adding N random images to the example images of a concept.

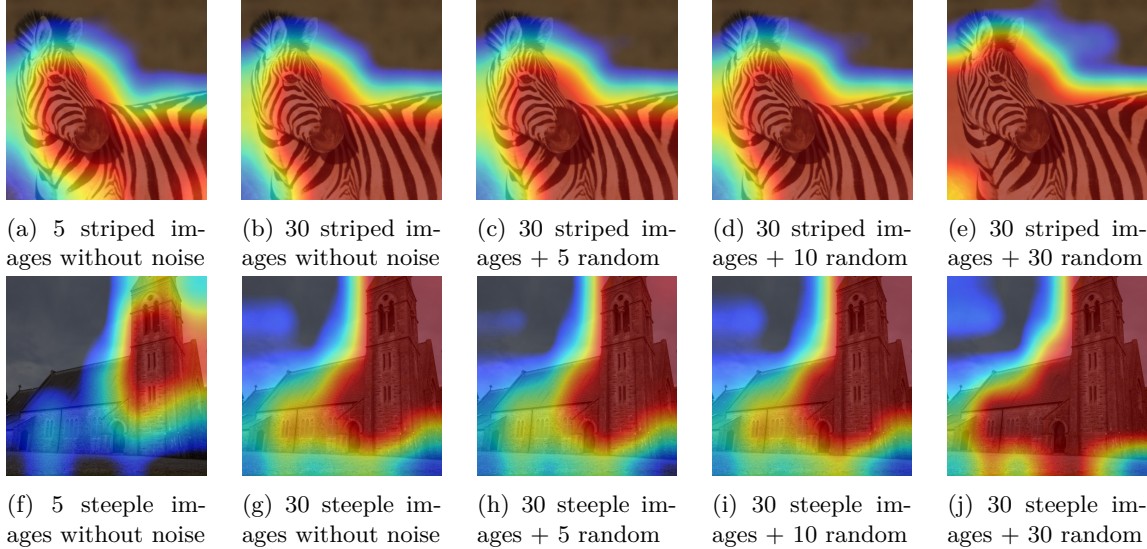

(a) 5 striped images without noise

(b) 30 striped images without noise

(c) 30 striped images + 5 random

(d) 30 striped images + 10 random

(e) 30 striped images + 30 random

(f) 5 steeple images without noise

(g) 30 steeple images without noise

(h) 30 steeple images + 5 random

(i) 30 steeple images + 10 random

(j) 30 steeple images + 30 random

Figure 19: Examples of concept maps computed with different amounts of example images and with different amounts of added noise via random images.

## H   Ablation study of concept map normalization

We normalize the raw concept map $M^c$ into $\hat{M}^c \in [0,1]$ using the bounded min–max approach in Eq. (2). Since this normalization depends on concept-specific bounds $(\ell_c, u_c)$ estimated from positive and random (negative) examples, we study how the choice of the summary statistic used to estimate these bounds affects the resulting concept maps, and we contrast it with a simple per-image min-max baseline.

The evaluation is carried out on the controlled tag validation setup from Section 4.4 using the same "100% tags" trained model. We consider the three tags $T, C, Z$ and the last three convolutional layers, whose spatial resolutions are $28\times28$, $14\times14$, and $7\times7$ respectively. For each tag, we use 200 positive test images (containing the tag) and 200 negative test images (same class images without the tag). Ground-truth tag masks are obtained deterministically from the tag generation procedure (using rectangle location and size) and then downsampled to the feature map grid of each layer, so that small discrepancies due only to upsampling do not influence localization scores. We compare four normalization variants. Three of them use the bounded min-max form in Eq. (2) and differ only in how the concept-level bounds $(\ell_c, u_c)$ are estimated from the concept and random sets. These are the following: *Chmean* (the default in the main paper) uses the contraharmonic-mean statistic, while *Max* and *Mean* replace it with the spatial maximum and spatial mean, respectively. In addition, we include a *Per-image min–max* baseline that normalizes each concept map independently using its own minimum and maximum, without concept-level bounds.

Localization quality on positive images is quantified using *Balanced MSE* (lower is better), which balances errors inside and outside the tag region, and *Soft IoU* (Rahman & Wang, 2016) (higher is better), a continuous, threshold-free version of the Jaccard index computed directly from the normalized concept map and the binary tag mask. To quantify how much and how often the concept map activates when it should not, we also report the *mean activation on negatives* (lower is better), defined as the spatial average of $\hat{M}^c$ on images where the tag is absent. Table 3 summarizes results averaged over the three layers. Results for $Z$ are reported for completeness but should not be used to evaluate normalization, since the "100% tags" model did not reliably learn the $Z$ tag and the corresponding localization scores are near random chance.

Table 3: Concept maps normalization ablation on the synthetic tag experiment. Results are averaged over the last three convolutional layers. Balanced MSE and Negative mean activation are better when lower, while Soft IoU is better when higher. Results for $Z$ are reported for completeness but should not be used to evaluate normalization since the underlying model did not reliably learn the $Z$ tag in this setting.

| Tag | Normalization | Balanced MSE↓ | Soft IoU↑ | Neg. mean act.↓ |
|-----|---------------|---------------|-----------|-----------------|
| C | Chmean | 0.0449 | 0.4252 | 0.0028 |
|   | Max | 0.0496 | **0.4601** | **0.0001** |
|   | Mean | 0.1055 | 0.2459 | 0.0337 |
|   | Per-image min–max | **0.0419** | 0.3995 | 0.4406 |
| T | Chmean | **0.0430** | **0.4460** | 0.0012 |
|   | Max | 0.0584 | 0.4449 | **0.0003** |
|   | Mean | 0.0982 | 0.2743 | 0.0169 |
|   | Per-image min–max | 0.0447 | 0.4139 | 0.2668 |
| Z | Chmean | 0.4892 | 0.0000 | 0.0000 |
|   | Max | 0.4892 | 0.0000 | 0.0000 |
|   | Mean | 0.4892 | 0.0000 | 0.0000 |
|   | Per-image min–max | 0.4904 | 0.0003 | 0.0062 |

On $T$ and $C$, *Chmean* yields the best (or near-best) Balanced MSE, suggesting concept maps that more closely match the desired "high inside and low outside" behavior. Compared to *Chmean*, the *Mean*-based variant consistently degrades both positive alignment and negative suppression, indicating that simple averaging is too sensitive to widespread low-magnitude activations. *Max* is consistently the most conservative choice on negatives (lowest mean activation), and can slightly improve Soft IoU for $C$, consistent with producing

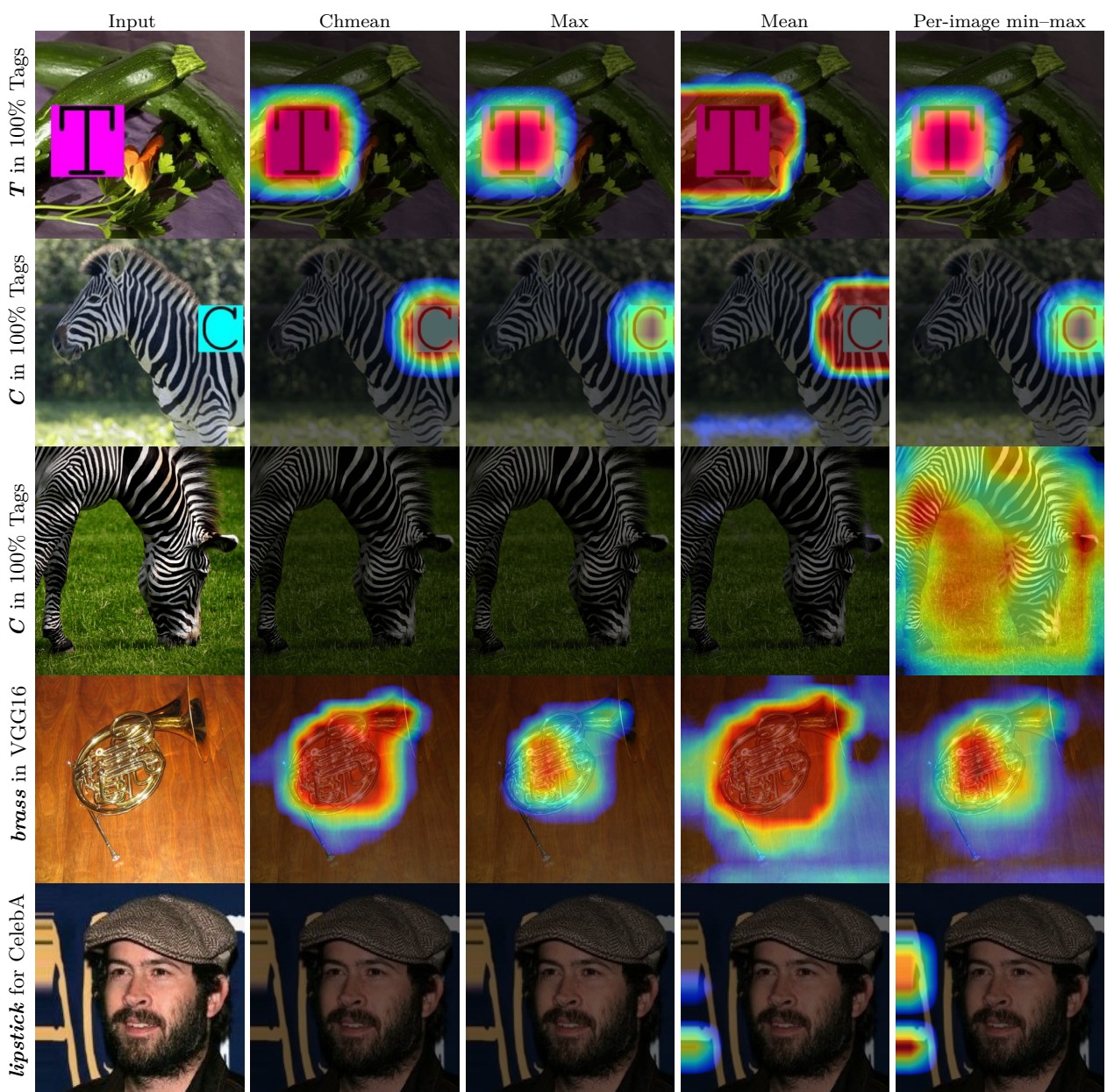

Figure 20: Qualitative comparison of concept map normalizations on synthetic tag examples and ImageNet and CelebA examples. Each row shows the same input image with overlays produced by different normalization strategies given a concept and a CNN model.

sparser maps. As a result, *Max* can be considered a reasonable and somewhat more strict alternative to *Chmean*. By contrast, *Per-image min–max* can appear competitive on positive alignment, but it substantially increases activation on negative images, highlighting the importance of concept-level bounds for meaningful cross-image comparisons. Qualitative examples on both the synthetic tag images and natural images are shown in Fig. 20, where each normalization is applied to the same inputs, and heatmaps are shown using only upscaling with disabled blur to better show the differences.

# I  IG baseline ablation

Our concept attribution is computed via Integrated Gradients (IG) in the feature maps space, i.e., by integrating gradients of the class logit w.r.t. the feature maps of a given layer. In the main paper, we implement IG using an all-zero feature map as a baseline (which we refer to as *Zero*). Since IG can depend on the baseline, in this appendix, we study how sensitive attributions are to changing such a baseline.

We compare *Zero* against two alternative baselines, which we refer to as *Mean* and *Random*. *Mean* is a dataset-mean baseline that uses, for each layer, the mean feature map computed from a pool of random images at that layer. *Random* is a random-noise baseline in feature maps space. For each test image, we sample Gaussian noise at the same spatial and channel resolution as the layer feature map, with noise scale matched to layer statistics estimated on the same random pool. The noise is centered at *Mean* but still enforcing non-negativity for post-ReLU layers. All results are reported as deviations w.r.t. the *Zero* baseline. The ablation is run using 5 concept-class pairs and the same 3 pretrained architectures (ResNet50V2, InceptionV3, and VGG16) used in the main paper, as well as the same 7 convolutional layers. The evaluated pairs are zebra (*striped*), honeycomb (*honeycombed*), crossword puzzle (*chequered*), waffle iron (*waffled*), and French horn (*brass*), and each pair uses 200 test images. As metrics, we report mean absolute error (MAE; lower is better) and Spearman rank correlation ($\rho$; higher is better).

Table 4: IG baseline ablation, summary by architecture. Values are deviations from the *Zero* baseline across 5 concept-class pairs and 7 layers for each architecture. Lower MAE is better, higher Spearman $\rho$ is better.

| Model | Baseline | MAE ↓ | Spearman $\rho$ ↑ |
|---|---|---|---|
| InceptionV3 | Mean | $0.0157 \pm 0.0150$ | $0.9802 \pm 0.0261$ |
| InceptionV3 | Random | $0.0272 \pm 0.0267$ | $0.9689 \pm 0.0520$ |
| ResNet50V2 | Mean | $0.0168 \pm 0.0205$ | $0.9882 \pm 0.0101$ |
| ResNet50V2 | Random | $0.0091 \pm 0.0084$ | $0.9795 \pm 0.0436$ |
| VGG16 | Mean | $0.0090 \pm 0.0098$ | $0.9870 \pm 0.0117$ |
| VGG16 | Random | $0.0233 \pm 0.0293$ | $0.9767 \pm 0.0243$ |

As shown in Table 4, both baselines show low MAE and high Spearman $\rho$ w.r.t. *Zero* across all architectures, indicating that Visual-TCAV attributions are largely robust to reasonable IG baseline choices. The dataset-mean baseline (*Mean*) is consistently close to *Zero* and exhibits stable rank agreement. The random-noise baseline (*Random*) remains strongly rank-consistent as well, while showing slightly larger absolute deviations and variability in some architectures (notably InceptionV3 and VGG16).

Table 5: Trend by layer depth across architectures and concepts. Values are deviations from *Zero*. Layers are aligned by a common depth index L1-L7 (from early to late). Results are aggregated across models and concept-class pairs. Lower MAE is better, higher Spearman $\rho$ is better.

| Layer | Mean | | Random | |
| | MAE ↓ | Spearman $\rho$ ↑ | MAE ↓ | Spearman $\rho$ ↑ |
|---|---|---|---|---|
| L1 | $0.0046 \pm 0.0046$ | $0.9872 \pm 0.0089$ | $0.0079 \pm 0.0066$ | $0.9792 \pm 0.0151$ |
| L2 | $0.0063 \pm 0.0074$ | $0.9850 \pm 0.0150$ | $0.0091 \pm 0.0096$ | $0.9719 \pm 0.0604$ |
| L3 | $0.0060 \pm 0.0056$ | $0.9891 \pm 0.0102$ | $0.0089 \pm 0.0089$ | $0.9833 \pm 0.0257$ |
| L4 | $0.0080 \pm 0.0053$ | $0.9797 \pm 0.0208$ | $0.0178 \pm 0.0166$ | $0.9703 \pm 0.0308$ |
| L5 | $0.0211 \pm 0.0234$ | $0.9887 \pm 0.0060$ | $0.0187 \pm 0.0143$ | $0.9858 \pm 0.0105$ |
| L6 | $0.0232 \pm 0.0177$ | $0.9803 \pm 0.0322$ | $0.0298 \pm 0.0295$ | $0.9640 \pm 0.0730$ |
| L7 | $0.0273 \pm 0.0172$ | $0.9860 \pm 0.0179$ | $0.0469 \pm 0.0395$ | $0.9708 \pm 0.0380$ |

Table 5 shows low MAE and high Spearman $\rho$ in all layers, with MAE that tends to increase slightly in deeper layers (L5-L7), while rank agreement remains generally high for both baselines. This is consistent with later layers typically producing larger-magnitude concept attributions, which can translate into a larger absolute MAE. The random-noise *Random* baseline is slightly less stable as it shows larger deviations than *Mean* in the deepest layers (notably L6-L7); however, it still largely preserves the overall ordering of images by concept attribution (high Spearman $\rho$).

## J  IG steps ablation

Our concept attribution is computed via IG in the feature maps space, i.e., by integrating gradients of the class logit w.r.t. the feature maps of a given layer. IG is approximated numerically using a finite number of integration steps, and the original IG paper suggests using between 20 and 300 steps (Sundararajan et al., 2017). In the main paper, we use 300 steps as a conservative choice for maximum fidelity; however, this also comes with a considerable computational cost. In this appendix, we study how sensitive attributions are to reducing the number of steps, comparing concept attributions between 20, 50, 100, and 200 steps against the 300 steps reference, reporting results as deviations w.r.t. 300 steps. The ablation is run using the same concept-class pairs, architectures, and layers used for the ablation baseline (Appendix I). Each pair uses 200 test images. As metrics, we report mean absolute error (MAE; lower is better) and Spearman rank correlation ($\rho$; higher is better).

Table 6: Summary by architecture of the IG steps ablation experiments. Values are deviations from the 300 steps setup and averaged across 5 concept-class pairs and 7 layers for each architecture, with runtime reported as average per concept-class pair. Lower MAE is better, higher Spearman $\rho$ is better. For comparison, the runtime at 300 steps for one concept-class pair is the following: ResNet50V2 = 6:32, InceptionV3 = 23:43, VGG16 = 16:04 (min:s).

| Model | Steps | MAE $\downarrow$ | Spearman $\rho \uparrow$ | Time (min:s) $\downarrow$ |
|---|---|---|---|---|
| InceptionV3 | 20 | $0.0002 \pm 0.0002$ | $0.9997 \pm 0.0004$ | 3:27 |
| InceptionV3 | 50 | $0.0001 \pm 0.0001$ | $0.9999 \pm 0.0001$ | 5:10 |
| InceptionV3 | 100 | $0.0000 \pm 0.0000$ | $1.0000 \pm 0.0000$ | 10:52 |
| InceptionV3 | 200 | $0.0000 \pm 0.0000$ | $1.0000 \pm 0.0000$ | 16:25 |
| ResNet50V2 | 20 | $0.0003 \pm 0.0003$ | $0.9999 \pm 0.0001$ | 2:16 |
| ResNet50V2 | 50 | $0.0001 \pm 0.0001$ | $0.9999 \pm 0.0001$ | 2:41 |
| ResNet50V2 | 100 | $0.0000 \pm 0.0000$ | $1.0000 \pm 0.0000$ | 3:16 |
| ResNet50V2 | 200 | $0.0000 \pm 0.0000$ | $1.0000 \pm 0.0000$ | 4:21 |
| VGG16 | 20 | $0.0019 \pm 0.0016$ | $0.9985 \pm 0.0013$ | 2:28 |
| VGG16 | 50 | $0.0006 \pm 0.0006$ | $0.9996 \pm 0.0004$ | 3:32 |
| VGG16 | 100 | $0.0003 \pm 0.0002$ | $0.9999 \pm 0.0002$ | 5:13 |
| VGG16 | 200 | $0.0001 \pm 0.0001$ | $1.0000 \pm 0.0001$ | 8:18 |

As shown in Table 6, reducing IG steps produces extremely small absolute deviations and near-perfect rank agreement w.r.t. the 300 steps reference across all architectures. Table 7 shows that stability is also consistent across layer depth. Even at 20 Steps, Spearman $\rho$ remains above 0.998 across models, while MAE remains below 0.002 (with the largest deviation observed for VGG16). At 50 steps, deviations further decrease, and rank agreement is essentially perfect with significantly less runtime.

Table 7: Trend by layer across architectures and concept-class pairs. Values are deviations from the 300 steps reference. Layers are aligned by a common depth index L1-L7 (from early to late). Lower MAE is better, higher Spearman $\rho$ is better.

| Layer | 20 Steps | | 50 Steps | | 100 Steps | | 200 Steps | |
|---|---|---|---|---|---|---|---|---|
| | MAE ↓ | Spearman $\rho$ ↑ | MAE ↓ | Spearman $\rho$ ↑ | MAE ↓ | Spearman $\rho$ ↑ | MAE ↓ | Spearman $\rho$ ↑ |
| L1 | $0.0008 \pm 0.0010$ | $0.9989 \pm 0.0010$ | $0.0003 \pm 0.0004$ | $0.9998 \pm 0.0003$ | $0.0001 \pm 0.0002$ | $0.9999 \pm 0.0001$ | $0.0000 \pm 0.0000$ | $1.0000 \pm 0.0000$ |
| L2 | $0.0009 \pm 0.0012$ | $0.9988 \pm 0.0017$ | $0.0003 \pm 0.0004$ | $0.9997 \pm 0.0004$ | $0.0001 \pm 0.0002$ | $0.9999 \pm 0.0001$ | $0.0000 \pm 0.0000$ | $1.0000 \pm 0.0000$ |
| L3 | $0.0007 \pm 0.0009$ | $0.9993 \pm 0.0008$ | $0.0002 \pm 0.0002$ | $0.9999 \pm 0.0001$ | $0.0001 \pm 0.0001$ | $0.9999 \pm 0.0000$ | $0.0000 \pm 0.0000$ | $1.0000 \pm 0.0000$ |
| L4 | $0.0011 \pm 0.0015$ | $0.9993 \pm 0.0008$ | $0.0004 \pm 0.0005$ | $0.9998 \pm 0.0002$ | $0.0002 \pm 0.0002$ | $0.9999 \pm 0.0001$ | $0.0000 \pm 0.0000$ | $1.0000 \pm 0.0000$ |
| L5 | $0.0006 \pm 0.0008$ | $0.9997 \pm 0.0004$ | $0.0002 \pm 0.0002$ | $0.9999 \pm 0.0001$ | $0.0001 \pm 0.0001$ | $0.9999 \pm 0.0001$ | $0.0000 \pm 0.0000$ | $1.0000 \pm 0.0001$ |
| L6 | $0.0007 \pm 0.0006$ | $0.9998 \pm 0.0002$ | $0.0002 \pm 0.0002$ | $0.9999 \pm 0.0001$ | $0.0001 \pm 0.0001$ | $1.0000 \pm 0.0000$ | $0.0000 \pm 0.0000$ | $1.0000 \pm 0.0000$ |
| L7 | $0.0008 \pm 0.0009$ | $0.9998 \pm 0.0003$ | $0.0003 \pm 0.0003$ | $0.9999 \pm 0.0001$ | $0.0001 \pm 0.0001$ | $1.0000 \pm 0.0001$ | $0.0000 \pm 0.0000$ | $1.0000 \pm 0.0000$ |

## K  C-Insertion and C-Deletion faithfulness experiment

This appendix provides a faithfulness experiment on larger models by intervening directly in feature-map space under the C-Insertion and C-Deletion frameworks. Given a concept direction, we progressively remove and re-insert its aligned component from the feature maps and measure the effect on the target class logit. We evaluate on all concept-class pairs analyzed in Figs. 5 and 12, for a total of 49 pairs, using ResNet50V2, InceptionV3, and VGG16 pre-trained on ImageNet, and the ResNet50V2 model trained on CelebA for gender classification. For each pair, we use the last layer and 200 test images.

For a given image, let $\mathbf{F}$ be the feature maps at the chosen layer, and let $\boldsymbol{p}^c$ be the pooled-CAV for concept $c$ (as defined in Section 3). At each spatial location $(i, j)$ we view $\mathbf{F}_{i,j,:}$ as a vector and define the following projection coefficient:

$$\alpha_{i,j} = \frac{\mathbf{F}_{i,j,:}^{\top} \boldsymbol{p}^c}{\|\boldsymbol{p}^c\|^2} \tag{7}$$

Let $\mathcal{D}_{\text{neg}}^c$ denote the set of negative (random) images associated with concept $c$ (i.e., the negative examples used when learning the CAV). We define $\beta$ as a baseline obtained by averaging $\alpha_{i,j}$ over all spatial locations and over all images in $\mathcal{D}_{\text{neg}}^c$. We then define the concept-aligned component at location $(i, j)$ as:

$$\mathbf{D}_{i,j,:}^c = ReLU\big(\alpha_{i,j} - \beta\big)\, \boldsymbol{p}^c \qquad \forall i, j \tag{8}$$

so that only the excess positive alignment with $\boldsymbol{p}^c$ relative to the negatives is removed.

We parameterize the intervention by $t \in [0, 1]$ and evaluate it on the uniform grid $t \in \{0, 0.1, \ldots, 1\}$, using a step size of 0.1. The C-Deletion and C-Insertion paths in the feature maps space are:

$$\mathbf{F}^{\text{del}}(t) = \mathbf{F} - t\, \mathbf{D}^c, \qquad \mathbf{F}^{\text{ins}}(t) = \mathbf{F} - (1 - t)\, \mathbf{D}^c. \tag{9}$$

C-Deletion starts from the original representation ($t=0$) and ends at a representation where the concept-aligned component is removed ($t=1$). C-Insertion starts from the removed representation ($t=0$) and returns to the original representation ($t=1$). For each $t$, we run a forward pass while replacing the feature maps with $\mathbf{F}^{\text{del}}(t)$ or $\mathbf{F}^{\text{ins}}(t)$, and record the target class logit, denoted by $z_k^{\text{del}}(t)$ and $z_k^{\text{ins}}(t)$. For post-ReLU layers, we apply a ReLU to the modified feature maps at each step. Since $\mathbf{F}^{\text{ins}}(t) = \mathbf{F}^{\text{del}}(1 - t)$, the C-Insertion trajectory is mathematically equivalent to the C-Deletion trajectory traversed in reverse. We still report C-Insertion curves for completeness w.r.t. the standard insertion and deletion terminology.

To summarize and aggregate data on each curve, we report an area under the curve (AUC) computed with the trapezoidal rule. For deletion, we integrate the logit curve directly, while for insertion, we integrate the logit gain relative to the removed baseline ($t=0$):

$$\text{AUC}_{\text{del}} = \int_0^1 z_k^{\text{del}}(t)\, dt, \qquad \text{AUC}_{\text{ins}} = \int_0^1 \big(z_k^{\text{ins}}(t) - z_k^{\text{ins}}(0)\big)\, dt. \tag{10}$$

Intuitively, a concept with a stronger positive influence on class $k$ should produce a lower $\text{AUC}_{\text{del}}$ (faster deletion) and a higher $\text{AUC}_{\text{ins}}$ (faster recovery). For reporting, we average AUCs across the 200 images of

Table 8: C-Deletion and C-Insertion summary by attribution rank. Values are aggregated over all evaluated concept-class pairs (200 images per pair). Lower $AUC_{del}$ and higher $AUC_{ins}$ indicate higher importance. We report Spearman $\rho$ between attribution rank and $AUC_{del}$, computed per class and averaged over classes.

| Model | Rank | $AUC_{del} \downarrow$ | $AUC_{ins} \uparrow$ | Spearman $\rho \uparrow$ |
|---|---|---|---|---|
| | Rank 1 | **14.23** | **4.23** | |
| ResNet50V2 (ImageNet) | Rank 2 | 18.37 | 0.49 | 1.00 |
| | Rank 3 | 18.87 | 0.02 | |
| | Rank 1 | **11.00** | **2.16** | |
| InceptionV3 (ImageNet) | Rank 2 | 12.10 | 1.12 | 0.75 |
| | Rank 3 | 12.96 | 0.27 | |
| | Rank 1 | **9.68** | **1.50** | |
| VGG16 (ImageNet) | Rank 2 | 10.32 | 0.91 | 1.00 |
| | Rank 3 | 10.75 | 0.46 | |
| ResNet50V2 (CelebA) | Rank 1 | **5.87** | **2.23** | 1.00 |
| | Rank 2 | 8.70 | -0.07 | |

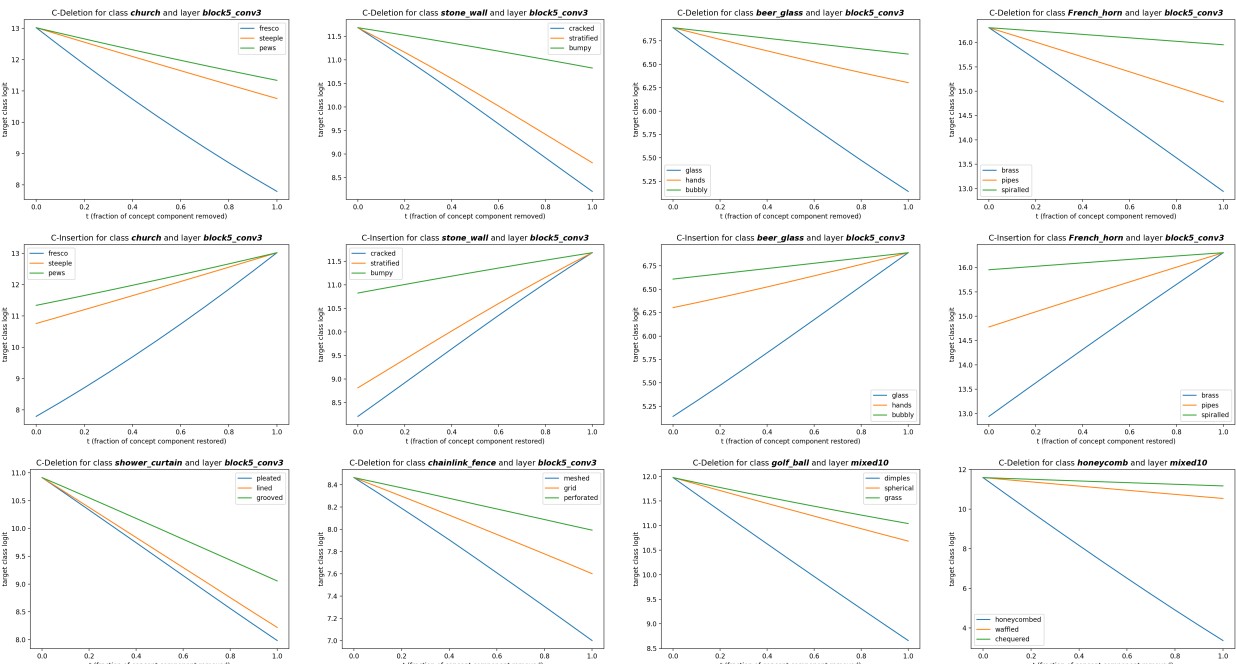

Figure 21: C-Deletion and C-Insertion curves. Each plot shows the mean target class logit over 200 images as a function of the deletion/insertion level $t$ (Part 1).

each pair, and aggregate results by model and concept rank (Rank 1 highest, Rank 3 lowest) using the global attribution ordering from Figs. 5 and 12 at the last layer.

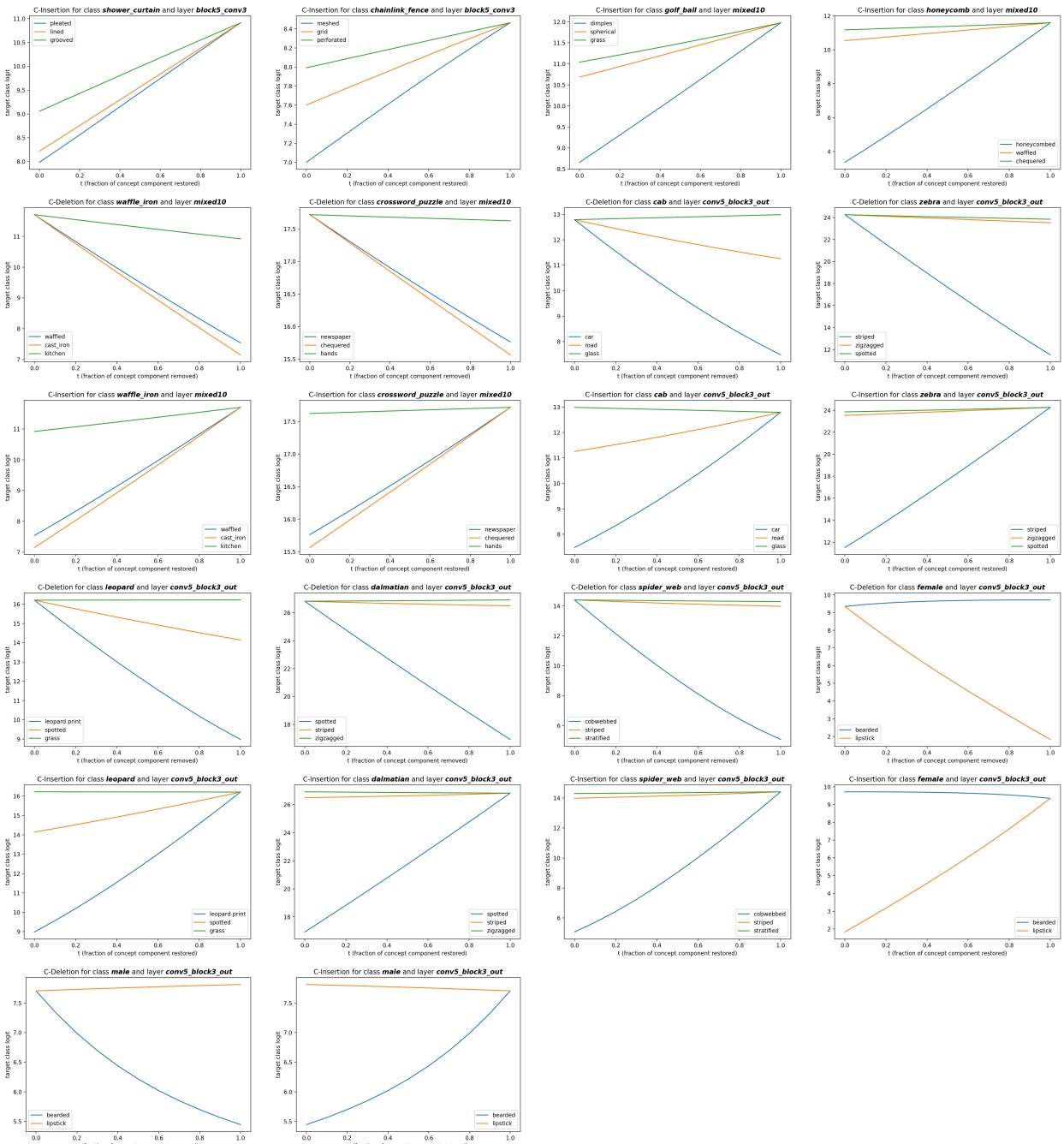

Figure 22: C-Deletion and C-Insertion curves. Each plot shows the mean target class logit over 200 images as a function of the deletion/insertion level $t$ (Part 2).

