# OpenReview forum: "Visual-TCAV: Concept-based Attribution and Saliency Maps for Post-hoc Explainability in Image Classification"
_TMLR — Accepted by TMLR_

### Review · Reviewer_yN7i · 2026-01-05

**Summary Of Contributions:**

The paper introduces Visual-TCAV, a novel explainability framework designed to enhance interpretability in Convolutional Neural Networks (CNNs). Visual-TCAV combines the strengths of state-of-the-art saliency and concept-based methods by offering both local and global explanations. It utilizes Concept Activation Vectors (CAVs) to generate class-agnostic saliency maps that highlight the regions of the input image where a certain concept is recognized by the network. Additionally, it estimates the contribution of these concepts to the network’s output using a generalization of Integrated Gradients, allowing for attribution of concepts in specific predictions. The framework is evaluated through a controlled experiment, demonstrating superior alignment with ground truth explanations compared to TCAV, thereby improving the faithfulness of concept attribution in CNN predictions.

**Audience:**

Yes

**Audience Explanation:**

The challenge focused in this paper along with the proposed idea provides insightful findings and guiding direction to the audience. However, more solid empirical evaluation and strong supportive evidence and analysis is needed.

**Broader Impact Concerns:**

There are no "Broader Impact Concerns" for this paper.

**Claims And Evidence:**

No

**Claims Explanation:**

# Strengths And Weaknesses

## Strengths
* The introduction of Visual-TCAV as a novel explainability framework bridges the gap between saliency-based and concept-based methods, offering a new solution to existing limitations in understanding how models make predictions.

* Visual-TCAV’s ability to generate visual explanations that show where the model identifies specific concepts within the input image adds a valuable layer of interpretability, addressing the lack of clarity in current saliency methods.

* By estimating the importance of human-defined concepts to specific model predictions, the framework enhances the transparency of CNNs, providing insight into how particular features contribute to decisions. The ability to apply Visual-TCAV for both local and global interpretability makes it versatile and applicable across a wide range of use cases, improving both the understanding of individual predictions and the overall behavior of the model.

## Weakness
* While the paper highlights the strengths of Visual-TCAV, it does not address potential limitations or challenges in applying the framework, such as computational complexity, scalability, or its effectiveness with different types of CNN architectures.

* Although the authors mention TCAV and other saliency-based approaches, there is no direct, quantitative comparison between Visual-TCAV and the latest methods in the field. A clear comparison would help better contextualize its strengths and weaknesses. Additionally, there is no solid large-scale analysis on the effectiveness of the proposed method.

* The effectiveness of Visual-TCAV is dependent on the user-defined concepts, which could introduce subjectivity or inconsistency in its application. This might affect its reliability in automatically identifying and attributing concepts without human intervention.

**Requested Changes:**

These are critical changes needed to improve the state of the paper.

1. It is insightful to present the sensitivity of the outcome of the proposed approach to the hyper-parameter choices and normalization method. In other words, some ablations studies on the decision choices are very needed.

2. It is necessary to measure using some evaluation metric on a statistically large dataset the performance of the proposed method as the current evidence is falling behind to strongly support the claim. So any additional quantitative results would be strongly recommended.

3. In section 3.3 it is missing references or explanation on the details of how IG is computed.

4. The related work has introduction contamination which are degrading the quality of the narration. It is advised to revise the two sections to properly structure the content.

non-critical requests are:
5. the Section 3.3 needs some minor rewording and restructuring to help the audience follow the details of the idea. It has some twist in the flow of the details at some points.

6. does the proposed idea bring extra computational overhead in comparison to the baseline methods?

---

> ### Author Response · Authors · 2026-02-20
> **Response to Reviewer yN7i**
>
> Thank you for taking the time to review our paper. We respond to the requested changes below.
>
> 1. We addressed this request by adding a set of ablation studies for key design choices and hyper-parameters. Specifically, we ablate the concept-map normalization (Appendix H), the Integrated Gradients baseline (Appendix I), and the number of IG integration steps (Appendix J), showing that the method’s importance rankings and concept maps remain largely stable to such choices.
>
> 2. We addressed this by adding a quantitative faithfulness evaluation based on concept deletion/insertion, aggregated over 49 concept–class pairs with 200 images each, showing that concepts ranked as more important by Visual-TCAV also induce stronger effects on the class logit if added or removed (Appendix K).
>
> 3&5. We addressed both comments by heavily revising Section 3.3. In particular, we restructured it into multiple paragraphs that explicitly separate the different steps (IG computation, attribution normalization, concept-weighting, spatial masking, global aggregation), and we added the missing IG reference together with a formal IG definition adapted to the feature map space. We hope this makes the section easier to follow, and we welcome further constructive feedback on this.
>
> 4. We addressed this comment by removing and compressing some sentences in the Related Work section that sounded a bit like motivation or problem statement.
>
> 6. Compared to TCAV, our pooled-CAV computation as "difference of means" is significantly cheaper than training many linear classifiers (e.g., SVMs). However, our attribution part is slower due to the use of IG instead of vanilla gradients. Computation time for IG heavily depends on the number of steps used. In the revision, we added Appendix J with an ablation study on the number of IG steps showing how stability and computational complexity change with 20, 50, 100, 200 and 300 steps.
>
> On the other weaknesses mentioned:
>
> Subjectivity in concept definition. To address this, we now explicitely mention the subjectivity point in the limitations. Still, we want to also note that in Appendix G, we evaluate robustness using different images for the same concepts and also to imperfect concept sets and both attributions and concept maps remain stable up to a 50/50 mix of real concept images and random ones, suggesting the method is not overly sensitive to concept set contamination.
>
> Comparison with other baselines. Here we assume (correct us if we are wrong) that you are referring to recent concept-discovery methods such as sparse autoencoders. However, these are not really comparable as their aim is to automatically extract a set of directions and interpret post-hoc what they encode, whereas Visual-TCAV is a concept-probing method that tests whether a user-specified concept is encoded as a meaningful direction in the model latent space and how it influences the prediction.
>
> We believe we have addressed all of your points, but if anything remains unclear or unconvincing, we would be happy to clarify. Thank you again for the constructive review.

---

### Review · Reviewer_Nb16 · 2026-01-05

**Summary Of Contributions:**

This paper introduces Visual-TCAV, a method that links together what a neural network focuses on and where those features appear in an image. It builds on TCAV by combining Concept Activation Vectors with a magnitude-based attribution method inspired by Integrated Gradients. first, it creates class-independent concept maps that highlight the spatial regions related to a concept, and second, it assigns a numerical score that reflects how much that concept influences the final class prediction.

**Strengths**
1. The paper addresses a usability gap in XAI by combining concept importance with spatial localization, providing more comprehensive diagnostics than Grad-CAM or TCAV alone.
2. The controlled tag experiment described in Section 4.4 establishes a quantitative benchmark by training models with synthetic tags at varying frequencies, thereby providing a ground-truth measure of explanation faithfulness.
3. The framework shows how concept representations develop across network layers, from low-level textures to higher-level objects, supporting systematic model analysis and debugging.

**Weaknesses**

1. The introduction section is not well developed. It does not clearly explain why global concept scores are insufficient in high-stakes domains. Explicit examples (e.g., medical imaging, where localization is critical) would strengthen the motivation.
2. The “interpretability gap” is described qualitatively. A formal statement framed in terms of unmet axiomatic properties would increase rigor.
3. Integrated Gradients uses a zero-filled baseline, despite prior work showing that fixed baselines can distort saliency maps. An ablation comparing alternative baselines is recommended.
4. The contraharmonic-mean-based normalization (Equation 2) is highly specific but not empirically validated. Comparisons with simpler statistics would clarify its benefit also show the theoretical justification.
5. ImageNet and CelebA results are mainly qualitative. Applying faithfulness metrics such as deletion/insertion would strengthen validation.
6. The 1–2 minute per-image runtime raises scalability concerns. A discussion of practical trade-offs and possible optimizations is needed.
7. Experiments are limited to CNNs. Even preliminary analysis or discussion of applicability to Vision Transformers would improve generalizability.
8. Most referenced studies are from 2016-2020, with limited engagement with recent XAI developments (2023-2024). The paper should compare against or discuss more contemporary concept-based methods, particularly recent work on automated concept discovery, multimodal explanations, and transformer-based interpretability techniques to demonstrate where Visual-TCAV stands in the current landscape.

**Audience:**

Yes

**Audience Explanation:**

Yes. The paper addresses a fundamental gap in explainable AI by unifying spatial localization with concept-based attribution, which is directly relevant to researchers working on model interpretability, trustworthy AI, and high-stakes applications like medical imaging where understanding both what and where the model recognizes is critical for practical deployment.

**Claims And Evidence:**

No

**Claims Explanation:**

Evidence for some key claims is not fully adequate.

Although the controlled tag experiment (Section 4.4) offers strong quantitative validation, some key design choices are not empirically justified. The use of a black baseline for IG and the contraharmonic mean normalization (Equation 2) are not supported by ablation studies. Moreover, the ImageNet and CelebA results rely mainly on qualitative visualizations rather than standard faithfulness metrics (e.g., deletion/insertion), leaving the accuracy of the concept maps only partially validated.

**Requested Changes:**

see weaknesses.

---

> ### Author Response · Authors · 2026-02-20
> **Response to Reviewer Nb16**
>
> Thank you for taking the time to review our paper. We respond to the weaknesses below.
>
> 1. We revised the Introduction to more explicitly motivate why global concept scores can be insufficient, expecially in high-stakes domains.
>
> 2. We addressed this by reframing our key contributions in the Introduction as three explicit desiderata (spatial grounding, per-instance attribution, and aggregatability)
>
> 3. We addressed this concern by adding an ablation on the IG baseline, comparing the zero baseline to mean and random baselines, and show that concept attributions are largely stable across baselines (see Appendix I).
>
> 4. We addressed this by adding a dedicated concept map normalization ablation that compares the contraharmonic-mean normalization (Eq. 2) against alternatives (i.e., mean, max and a simple per-image min–max), and we report quantitative localization metrics on a controlled ground-truth setting, as well as qualitative examples (Appendix H).
>
> 5. We addressed this by adding a faithfulness evaluation based on concept deletion/insertion, showing that concepts that our attributions deems more important also induce stronger effects on the class logit on both ImageNet and CelebA (Appendix K).
>
> 6. We clarify that the runtime reported in the submission corresponds to computing IG with 300 integration steps, which we used to generate the paper results with maximum numerical precision. For practical scalability, the number of steps can be reduced substantially while still providing a very accurate approximation of the IG integral. We added Appendix J, where we evaluate 20/50/100/200 steps and show that attributions remain highly consistent with the 300 steps reference while reducing runtime by up to 8x. Furthermore, 1-2minutes for a local explanation (with 7 layers and 300 IG steps) does not scale linearly because when running Visual-TCAV for only one image there is also a significant overhead for computing the CAV. For instance, the time to run 200 images with 7 layers and 300 IG steps would take somewhere around 15 minutes (or 2-3 minutes with 20 IG steps).
>
> 7. We addressed this by expanding the Limitations and Future Work section with a discussion on the applicability of Visual-TCAV to Vision Transformers, explaining why spatial grounding is challenging due to the fact that patch tokens can encode evidence originating from other regions as a result of token mixing in self-attention. This could make concept maps potentially misleading and addressing it would require non-trivial design adaptations, therefore we leave extension to ViT as a future standalone work.
>
> 8. In our related work, we already cite and discuss recent advancements in concept discovery, as we believe these works are very important and adjacent to ours. In particular, one of the most promising approach that has been shown to work for both transformers and multimodal models are sparse autoencoders (SAEs). The aim of these methods is to automatically extract a set of meaningful concept directions from a model, and then trying to interpret post-hoc what they encode. Visual-TCAV, on the other hand, allows an user to test whether a concept of interest is encoded in the model as a meaningful direction and test how it influences the prediction. In other words, these two approaches (probing vs discovering) are probably better seen as different tools in the intepretability analyst toolbox to be used for different purposes rather than as alternatives. Furthermore, recent findings have shown that often directions that encode useful concepts of interest do not even exist among SAEs neurons and it's unclear what causes this problem [1]. Without an "ultimate concept discovery method" that can comprehensively extract all concepts from a model, concept-based probing is (in our view) likely to remain a relevant and useful tool.
>
> [1] Sharkey et al, Open Problems in Mechanistic Interpretability, TMLR, 2025
>
> We believe we have addressed all of your points, but if anything remains unclear or unconvincing, we would be happy to clarify. Thank you again for the constructive review.

---

### Review · Reviewer_1JN7 · 2026-02-10

**Summary Of Contributions:**

The paper proposes Visual-TCAV, an explainability framework that combines concept-based explanations with traditional saliency maps to better interpret CNN image classifiers. In simple terms, the authors extend the popular TCAV method (Testing with Concept Activation Vectors) by not only indicating whether a high-level concept influences a class prediction, but also where and how much that concept contributed to a specific image. This bridges the gap between global explainability (concept importance for a class overall) and local explainability (visualising features for a single image). The method can be applied to any convolutional layer of a CNN.

Key contributions:

* Visual Concept Localisation: The paper proposes concept-saliency maps that highlight the parts of an input image the model associates with a user-defined concept. These maps can highlight where the model detects concepts such as stripes or wheels in a given image.
* Concept Attribution to Predictions: The framework uses generalised Integrated Gradients to calculate how much a concept affects the model’s output for a certain class in a given example. Unlike the original TCAV, which only indicated sensitivity qualitatively or via a significance test, Visual-TCAV provides a detailed, per-instance score.
* Unified Local and Global Explanations: The framework supports both local explanations (for individual predictions) and global explanations (aggregated insights over a dataset or class). The user can either generate concept-attribution heatmaps for a single image or compute the concept’s average contribution for a class across images, for example, analysing whether the concept grass is generally important for the class golf ball across the dataset. Visual-TCAV can hence explain a single prediction at a time and also summarise model behaviour across the dataset, addressing one of the key limitations that saliency maps alone cannot handle.



Strengths. The paper is well motivated. It identifies clear limitations of existing methods: saliency maps cannot tell us which high-level feature drove a prediction, while TCAV cannot localise concepts or measure per-instance importance. As concept-based interpretability has attracted much interest in recent years, the proposed solution is novel and timely. In addition, the authors design experiments to demonstrate how well the method's output aligns with ground-truth annotations, and provide several quantitative examples. They also demonstrate the stability of the explanations produced by their method to noise and illustrate that it is model-agnostic and can be applied across layers of the model.

Weaknesses: One key limitation of TCAV is the need for user-defined concept examples, which Visual TCAV also inherits. This would mean the method's performance relies heavily on the example images the user chooses to represent the concepts. Another weakness is the method's computational overhead: computing Integrated Gradients for each concept can be expensive in practice. In addition, the authors do not directly compare with other concept-based localisation techniques that do address the limitations of TCAV. The method also assumes that the concepts are tested one at a time, and if multiple concepts overlap in an image, their attributions are treated separately. None of these weakness are critical, though I would encourage the authors to clarify the scope and advantages of Visual TCAV

**Audience:**

Yes

**Audience Explanation:**

The submission would be of interest to TMLR’s audience, particularly the explainable AI community. The proposed framework aims to address the limitations of two popular explainability methods: TCAV and saliency maps. For an end user, it is particularly useful to visualise and understand the inner workings of a black-box Deep Neural Network.

**Broader Impact Concerns:**

I do not see any serious ethical or broader-impact issues with this work.

**Claims And Evidence:**

Yes

**Claims Explanation:**

In general, the claims made in the paper are well supported by both quantitative and qualitative evidence. To validate their core claim that their method can identify and attribute concepts more reliably than existing methods, the authors carefully design an experiment in which they train multiple CNNs on a crafted dataset in which images are tagged with spurious concepts, and demonstrate their method correctly identifies important concepts the majority of the time, as opposed to the existing methods. The significant gap supports the claim that their method produces faithful explanations.

To demonstrate that Visual TCAV can localise concepts within an image, the authors provide multiple figures with concept heatmaps. These quantitative results support the claim that their approach localises where the model sees the concept. In addition, to illustrate that their method can generate global explanations, the authors also aggregate concept attributions across many images to produce a class-level summary.

**Requested Changes:**

I would encourage the authors to compare their methods, such as CAVLI (Shukla et al., 2023), or other unsupervised concept-discovery approaches. It would strengthen the submission if the authors could add a brief quantitative and qualitative comparison to CAVLI.

I would also recommend that the authors explicitly clarify the scope and the limitations of the proposed approach. For instance, if a user misses out on adding a crucial concept, the method won’t explain the model's reliance on it. The paper alludes to the non-additivity of concept attributions. I would ask the authors to explicitly call it out.

In addition, it would help the reader if the authors explicitly explained the intuition behind their design choices; for instance, they use a custom normalisation of the concept maps and a ReLU activation on the pooled CAV.

The authors should also add a note regarding the computational cost of the framework.  It would strengthen the paper to include a brief note on runtime or complexity, e.g., how long it takes to produce a concept attribution map for a single image relative to a standard Grad-CAM.

---

> ### Author Response · Authors · 2026-02-20
> **Response to Reviewer 1JN7**
>
> Thank you for taking the time to review our paper. We respond to the requested changes below.
>
> 1.  We were not able to find a publicly available implementation of CAVLI at the time of revision. Still, the CAVLI paper notes that their CDS scores resemble the TCAV score, for which we provide a comparison. Regarding unsupervised concept-discovery methods (e.g., sparse autoencoders), these are not really comparable as their aim is to automatically extract a set of directions and interpret post-hoc what they encode, whereas Visual-TCAV is a concept-probing method that tests whether a user-specified concept is encoded as a meaningful direction in the model latent space and how it influences the prediction.
>
> 2. We addressed this comment by better clarifying the non-additivity at the end of Section 3.3. We also expanded the Limitations to state the practical following implication: since concept attributions are not additive, Visual-TCAV does not provide a completeness guarantee over an arbitrary user-defined concept set, and therefore cannot verify that the tested concepts are exhaustive of what the model has learned.
> Still, the lack of an additivity guarantee does not necessarily mean that we do not have diagnostic tools that can give us an hint on whether we are missing crucial concepts or not. For instance, in the revision, we added a faithfulness evaluation based on C-Deletion/C-Insertion (Appendix K), where we intervene by removing concept-aligned components from the feature maps. If deleting a set of tested concepts produces only a limited change in the target logit and does not flip the prediction, this could be interpreted as "maybe we are missing something important".
>
> 3. We addressed this by adding a set of ablation studies for key design choices. Specifically, we ablate the concept-map normalization (Appendix H), the Integrated Gradients baseline (Appendix I), and the number of IG integration steps (Appendix J), showing that the method’s importance rankings and concept maps remain largely stable to such choices. On the ReLU specifically, it is done because the do not want to differentiate the regions of the image between negatively aligned with the concept and orthogonal, but we only want to highlight the positively aligned ones. This is also the same reason Grad-CAM equation has a ReLU. Ours is analogous but with the pooled-CAV instead of the global-average-pooled gradients.
>
> 4. While the submission already included a discussion on runtime, we now added an Appendix J, where, since the number of IG steps are the main driver of computational cost, we evaluate 20/50/100/200 steps and show that attributions remain highly consistent while reducing runtime by up to 8x. Furthermore, considering TCAV as a baseline, our pooled-CAV computation as "difference of means" is significantly cheaper than training many linear classifiers (e.g., SVMs). However, our attribution part is slower due to the use of IG instead of vanilla gradients. Compared to IG as baseline, complexity is similar because the bottleneck is the same, while Grad-CAM would be the fastest since it only requires to backpropagate gradients once on top of the forward pass.
>
> We believe we have addressed all of your questions, but if anything remains unclear or unconvincing, we would be happy to clarify. Thank you again for the constructive review.

---

### Decision · Action_Editor_97Fq · 2026-03-30

**Recommendation:** Accept as is

**Audience:**

Yes

**Audience Explanation:**

The paper provides a new tool for interpreting image classification with CNNs. A paper that many ML and CV researchers are interested in.

**Claims And Evidence:**

Yes

**Claims Explanation:**

The reviewers value the combination of the TCAV technique for explainability and the visual representation, which together improve the interpretability of concepts in images. The reviewers provided a few pointers for improvement, and the authors have mostly responded positively to the reviewers' requests. Computational complexity is still a salient issue; however, it is not a limiting one.

The authors provide their code as a zip file and mention in their response that they could not compare it with CAVLI because there is no public repository for that solution. I ask the authors to add a GitHub, Hugging Face, or other public repository for their code and include that link in their paper. Please add the link to the abstract. Given that this is the only change I request, I recommend Accept as is. However, I will check before publishing.